

# Survey on graph embeddings and their applications to machine learning problems on graphs

Ilya Makarov[1,2], Dmitrii Kiselev[1], Nikita Nikitinsky[3] and Lovro Subelj[2]

[1] HSE University, Moscow, Russia
[2] Faculty of Computer and Information Science, University of Ljubljana, Ljubljana, Slovenia
[3] Big Data Research Center, National University of Science and Technology MISIS, Moscow, Russia

Corresponding authors
Ilya Makarov, iamakarov@hse.ru
Dmitrii Kiselev, dkiseljov@hse.ru

## ABSTRACT

Dealing with relational data always required significant computational resources, domain expertise and task-dependent feature engineering to incorporate structural information into a predictive model. Nowadays, a family of automated graph feature engineering techniques has been proposed in different streams of literature. So-called graph embeddings provide a powerful tool to construct vectorized feature spaces for graphs and their components, such as nodes, edges and subgraphs under preserving inner graph properties. Using the constructed feature spaces, many machine learning problems on graphs can be solved via standard frameworks suitable for vectorized feature representation. Our survey aims to describe the core concepts of graph embeddings and provide several taxonomies for their description. First, we start with the methodological approach and extract three types of graph embedding models based on matrix factorization, random-walks and deep learning approaches. Next, we describe how different types of networks impact the ability of models to incorporate structural and attributed data into a unified embedding. Going further, we perform a thorough evaluation of graph embedding applications to machine learning problems on graphs, among which are node classification, link prediction, clustering, visualization, compression, and a family of the whole graph embedding algorithms suitable for graph classification, similarity and alignment problems. Finally, we overview the existing applications of graph embeddings to computer science domains, formulate open problems and provide experiment results, explaining how different networks properties result in graph embeddings quality in the four classic machine learning problems on graphs, such as node classification, link prediction, clustering and graph visualization. As a result, our survey covers a new rapidly growing field of network feature engineering, presents an in-depth analysis of models based on network types, and overviews a wide range of applications to machine learning problems on graphs.

## INTRODUCTION

Many instances in the real world can be modeled as graphs or networks. Some of the typical examples include social interactions, biological data, such as protein interactions or neural connections, links between websites on the Internet, etc. One of the main goals of graph modeling is to formulate a general technique capable of processing structural data including relations between objects, which may also have some domain-specific information. For example, given a social network, we might be interested in predicting whether a pair of users are friends, or in identifying communities of interconnected users. The former leads to a link prediction problem on the graph, while the latter describes a node clustering problem.

We focus on graph representation theory, aiming to automatically learn low-dimensional vector features for the simplest graph motifs, such as nodes and edges, in a way that would enable efficiently solve machine learning problems on graphs including node classification, link prediction, node clustering, while also tackling approaches for graph similarity and classification, and general aspects of graph visualization.

Before the emergence of the area, the extraction of important features for predictive tasks on graphs had to be manually engineered. It required a lot of efforts from the domain experts. For example, many approaches for graph representation rely on extracting summary statistics, such as vertex degrees or clustering coefficients (*Bhagat, Cormode & Muthukrishnan, 2011*) popular in social sciences, graph kernels (*Vishwanathan et al., 2010*) particularly used in computational biology to compute inner product similarities between graphs, or specifically designed features to measure neighborhood similarity (*Liben-Nowell & Kleinberg, 2007*). In addition to the time-consuming feature engineering, such summaries were very inflexible, task/data-dependent, and did not generalize well across different prediction tasks on graphs. An alternative methodology is to learn feature representations automatically as an optimization problem. The goal is to design objective cost functions that capture dependencies and similarities in a graph while preserving high quality in relational machine learning tasks and constructing graph embeddings under efficiency constraints over time and memory.

Today, there exists a large variety of graph embeddings automatically extract vector representation for networks (*Moyano, 2017*; *Hamilton, Ying & Leskovec, 2017b*; *Cai, Zheng & Chang, 2017*; *Cui et al., 2018*; *Goyal & Ferrara, 2017*; *Chen et al., 2018a*; *Wu et al., 2019b*), knowledge graphs (*Nickel et al., 2016*) and biological data (*Su et al., 2020*). Some of these algorithms only work with structural information, such as popular Node2vec (*Grover & Leskovec, 2016*), LINE (*Tang et al., 2015*), DeepWalk (*Perozzi, Al-Rfou & Skiena, 2014*), while others like GCN (*Kipf & Welling, 2016a*), GraphSAGE (*Hamilton, Ying & Leskovec, 2017a*), VGAE (*Kipf & Welling, 2016b*) also use node attributes. The methods also differ based on whether a given graph is (un)directed, (un)weighted, (non-)attributed, (dis)assortative, if it changes over time in terms of adding/deleting nodes/edges, and whether they use a transductive or inductive approach for learning network dynamics inference. All of these models have their advantages and shortcomings, but what unifies them is the unique pipeline to verify the network embedding model in terms of the quality

of machine learning tasks on benchmark datasets. In addition, authors measure construction and inference time efficiency, memory consumption, and a possibility to include graph dynamics in the model.

Most surveys on graph embeddings provide a simple taxonomy for graph models based on how the model is fitted and only show applications within the graph domain, for example, node classification or link prediction (*Moyano, 2017*; *Hamilton, Ying & Leskovec, 2017b*). *Goyal & Ferrara (2017)* provide experiments and study the influence of hyperparameters on different tasks. Some works focus on a specific field such as attention models (*Lee et al., 2019*) and graph neural networks (*Wu et al., 2019b*; *Chen et al., 2018a*; *Zhang, Cui & Zhu, 2018*). *Cui et al. (2018)* compare models in terms of what information they preserve: structure and properties or side information. Neural network approaches are usually classified by the core architecture, for example, recurrent neural networks (RNN) or convolutional neural networks (CNN), and losses for different tasks, such as cross-entropy for link prediction and node classification and reconstruction loss for unsupervised representation learning. *Chen et al. (2018a)* provides meta-strategies for choosing embedding models, but examine only deep learning based methods. *Lee et al. (2019)* follow the classification of *Cai, Zheng & Chang (2017)* and separate attention models by type of input and output, deriving recommendations for working with different graphs (heterogeneity, multi-view, directed acyclic graphs) and on different tasks (node classification, clustering, ranking, alignment, link prediction). *Zhang, Cui & Zhu (2018)* is quite similar to other GNN surveys, but also provides an overview of modern models and tasks like reinforcement learning on graphs, analyses techniques for better representation learning like sampling strategies, skip connections, inductive learning and adversarial training.

In contrast, our work tries to generalize the advances of previous surveys. Our survey is not limited to specific model types and provides an overview from different angles: training process, input graph properties, specific tasks and applications in a non-graph domain, and open problems, etc.

The paper is structured as follows. We start with a brief explanation of general approaches to learn network embedding and introduce to a reader the core ideas of graph representation models. Next, we describe different models adapted to specific types of networks. Then, we state the most crucial machine learning problems on graphs and solutions to them based on network embeddings. To cover the use of overviewed models, we provide applications to other machine learning domains. We finalize review sections with the listing of open problems in the field of network representation learning.

Finally, we provide our experiments to understand in practice, how different graph embeddings perform on benchmark network datasets and interpret, why the chosen graph embedding model with a given training setting result in good or bad quality on a given benchmark dataset and how it is related to the method behind the model. Our experiment section aims to show how one can choose the best graph embedding by the nature of the model construction and network descriptive statistics, which is one the most interesting problems for practical applications of graph embeddings for machine learning frameworks.

## PRELIMINARIES

Before describing any methods we need to introduce some definitions. We will use $V$ as a set of graph vertices, $E$ as a set of graph edges, $A$ as graph adjacency matrix and $G(V, E)$ as graph description. The procedure on constructing vector representation of a graph we are interested in is called graph embedding.

**Definition 1 (Graph embedding)** *is a mapping from a collection of substructures (most commonly either all nodes, or all edges, or certain subgraphs) to $\mathbb{R}^d$. We will mostly consider **node embeddings**: $f : V \to \mathbb{R}^d,\ d \ll |V|$.*

For many graph-based tasks, the most natural task formulation is unsupervised learning: this is the case when we need to learn embeddings using only the adjacency matrix $A$ containing information on structural similarity and possibly **attributed features** $X$, but without task-specific loss part. It is also possible that there are labels available for some substructures of the graph, and we wish to recover missing labels in a semi-supervised approach. One example of this is node classification, in which all nodes are available from the outset, but only a fraction is labeled.

Now let us clarify what is meant by a good embedding. By the embedding procedure, one should aim to compress the data, while retaining most of the essential information about similarities and simultaneously, extract important features from the structural information. What counts as essential may vary depending on an intended application; most common properties we want to capture in a graph are termed as **node proximity** and **structural similarity** (neighbourhood information and structural role, respectively).

**Definition 2 (First and second order proximities)** *The first-order proximity describes the pairwise proximity between vertices. For any vertices, the weight $a_{ij}$ (possibly zero) of the edge between $v_i$ and $v_j$ characterizes the first-order proximity between these vertices, thus representing adjacency matrix $A = (a_{ij})_{i,j=1}^{n}$. A neighborhood of vertex $v_i$ is defined as a set of adjacent vertices $N_{v_i} = \{v_k | a_{ik} > 0,\ k \neq i\}$ thus meaning that vertex itself is not included in its neighborhood. The second-order proximity between a pair of vertices $v_i$ and $v_j$ describes the similarity measure between their neighborhood structures $Nv_i$ and $Nv_j$ with respect to a selected proximity measure.*

## METHODS FOR CONSTRUCTING GRAPH EMBEDDING

We briefly describe graph embedding methods of three general categories, corresponding to the perspective they take on embedding graphs: matrix factorizations, node sequence methods and deep learning based methods. These are, of course, not mutually exclusive, but it is more convenient to adhere to their primary features. We also cover a specific type of embeddings based on embedding space metric.

We select papers from several curated lists and major conferences on network science, artificial intelligence, machine learning and data mining, as well as core research publishers and indexing services. Paper sources are referred in Table 1. We used the following keywords: graph/network embeddings, graph/network representation, graph neural networks, graph convolutional networks, graph convolution, graph attention,

**Table 1  Paper sources.**

| Name | Link | Description |
|---|---|---|
| Curated lists | | |
| by Chen | https://github.com/chihming/ | awesome-network-embedding |
| by Rozemberczki | https://github.com/benedekrozemberczki/ | awesome-graph-classification |
| by Rebo | https://github.com/MaxwellRebo/ | awesome-2vec |
| by Soru | https://gist.github.com/mommi84/ | awesome-kge |
| Conferences | | |
| Complex Networks | https://complexnetworks.org/ | International Conference on Complex Networks and their Applications |
| The Web | https://www2020.thewebconf.org/ | The Web Conference is international conference on the World Wide Web. |
| WSDM | http://www.wsdm-conference.org/ | Web-inspired research involving search and data mining |
| IJCAI | https://www.ijcai.org/ | International Joint Conferences on Artificial Intelligence |
| AAAI | https://www.aaai.org/ | Association for the Advancement of Artificial Intelligence |
| ICML | https://icml.cc/ | International Conference on Machine Learning |
| SIGKDD | https://www.kdd.org/ | Special Interest Group in Knowledge Discovery and Databases |
| Domain conferences | | |
| ACL | http://www.acl2019.org/ | Association for Computational Linguistics |
| CVPR | http://cvpr2019.thecvf.com/ | Conference on Computer Vision and Pattern Recognition |
| Publishers | | |
| ACM DL | https://dl.acm.org/ | Full-text articles database by Association for Computing Machinery |
| IEEE Xplore | https://ieeexplore.ieee.org/Xplore/home.jsp | Research published by Institute of Electrical and Electronics Engineers |
| Link Springer | https://link.springer.com/ | Online collection of scientific journals, books and reference works |
| Indexing services | | |
| Scopus | https://www.scopus.com/ | Abstract and citation database |
| Web of Science | https://www.webofknowledge.com/ | Citation Indexer |
| Scholar Google | https://scholar.google.com/ | Web search engine for indexing full-text papers or its metadata |

graph/network classification/link prediction/clustering, deep learning for graphs, geometric deep learning, GCN, GNN, GAT.

Historically the first graph embedding methods were factorization based, which generally try to approximate a large matrix with a low-rank matrix factorized into a product of two matrices containing representations, thus modeling each entry of the original matrix with an inner product of representations. Sequence-based embeddings linearize the graph using random walks or diffusion and maximize the probability of observing the neighborhood (context) of a node given its embedding. Deep learning-based models learn a function mapping a graph in the numeric form to a low-dimensional embedding by optimizing over a broad class of expressive neural network functions.

## Dimensionality reduction (matrix factorization) methods

**Definition 3 (Matrix factorization)** *is a decomposition of a matrix to the product of matrices. In this sense, the first matrix in series is named self node representation and the last matrix refers to node context.*

Factorization models are common techniques in different machine learning domains to receive meaningful low-dimensional representation. Moreover, a lot of methods use similarity matrix between observations, which can also be reformulated as the graph similarity matrix.

Factorization techniques can be applied to a different graph representations and optimize different objectives. Some methods directly decompose the adjacency matrix $A$, for example, MDS (*Kruskal & Wish, 1978*) reconstructs it by minimizing MSE between element $a_{ij}$ and euclidean distance between vectors $u_i$ and $u_j$ of manifold $U$. We can rewrite this with expression $\sum_{i=1}^{N} \sum_{j=1}^{N} \left( a_{ij} - \|u_i - u_j\|_2^2 \right)^2$. LSI (*Deerwester et al., 1990*) simply applies singular value decomposition to $A$ *Golub & Reinsch (1971)*. In *Wold, Esbensen & Geladi (1987)* the manifolds are learned by maximizing variance for linear mixture. It is extended by LDA *Martinez & Kak (2001)*.

Another way to use dimensionality reduction is to build proximity matrix of the graph. For example, IsoMap (*Tenenbaum, De Silva & Langford, 2000*) use shortest path matrix $D$ and apply MDS to learn embeddings. LLE (*Roweis & Saul, 2000*) learns node similarity by reconstructing weights matrix $W$ with which neighboring nodes affect each other: $\|X - W^T U\|_2^2$ and repeats that procedure to learn manifold $U$ with achieved matrix $W$. LPP (*He & Niyogi, 2004*) estimates the weighted matrix $W$ as heat kernel and learn manifold $U$ by reduction of $W$ with Laplacian Eigenmaps technique. IsoMap and LLE were proposed to model global structure while preserving local distances or sampling from the local neighborhood of nodes. The lower bound for methods complexity was quadratic in the number of vertices, still making them inappropriate for large networks.

**Definition 4 (Graph Laplacian)** If matrix D is the diagonal degree matrix, that is $D = \text{diag}(\Sigma_j A_{ij})$, then Laplacian matrix can be defined as L = D − A.

Another approach for spectral graph clustering (*Chung & Graham, 1997*) was suggested in *Belkin & Niyogi (2002)* named Laplacian eigenmaps (LE), representing each node by graph Laplacian eigenvectors associated with its first $k$ nontrivial eigenvalues. The goal for Laplacian Eigenmaps class of models lies in preserving first-order similarities. Thus, a model gives a larger penalty using graph Laplacian if two nodes with larger similarity are embedded far apart in the embedding space. Laplacian objective function is symmetric in each pair $(i, j)$, and thus it cannot capture edge orientations. Kernel Eigenmaps (*Brand, 2003*) extends this approach to nonlinear cases. In contrast to LE, which preserved nodes dissimilarity, Cauchy embedding (*Luo et al., 2011*) proposes optimization condition modification which preserves the similarity between vertices. Structure Preserving Embedding (SPE) (*Shaw & Jebara, 2009*) aims to use LE combined with preserving spectral decomposition representing the cluster structure of the graph. It introduces a new graph kernel and applies SVD to it.

Graph Factorization (GF) (*Ahmed et al., 2013*) try to solve the scalability issue of factorization methods by decreasing node neighborhood via graph partitioning and utilizing distributed computation.

The models in this class can be either symmetric and obtain final representations only from embedding matrix. GraRep (*Cao, Lu & Xu, 2015*) consider $k$-hop neighborhood ($A^k$)

using SVD decomposition of $A^k$. HOPE (*Ou et al., 2016*) is specific asymmetric transitivity preserving graph embedding. It is found that most asymmetric similarity measures can be formulated as $S = M_g^{-1} M_l$. Katz index refers to $M_g = I - \beta A$, $M_l = \beta A$. Rooted PageRank can be stated as $M_g = I - \alpha P$, $M_l = (1 - \alpha) P$. Common neighbors is represented by $M_g = I$, $M_l = A^2$, and Adamic-Adar with $M_g = I$, $M_l = A \cdot D \cdot A$. To avoid calculation of similarity matrix authors propose to use generalized SVD and directly estimate matrices $M_g$ and $M_l$. *Abu-El-Haija, Perozzi & Al-Rfou (2017)* proposed to use concatenation of two node representations capturing in- and out-connections. Authors of *Wang et al. (2017d)* proposed a Modularized Nonnegative Matrix Factorization (M-NMF) model to preserve the community structure in network representation. In ATP model (*Sun et al., 2019*) authors embed directed graph constructing two vectors for each node via factorization framework. *Kefato, Sheikh & Montresor (2020)* propose multi-objective framework for preserving directed nature of graph. SDNE (*Wang, Cui & Zhu, 2016*) uses autoencoders (as neural network based dimension reduction technique) to capture non-linear dependencies in local proximity.

Factorization based models are the best-studied theoretically and provide a well-known general framework for graph embedding optimization (*Liu et al., 2019*), however, they suffer from high computational complexity for large graphs and often capture only a small-order proximity *Perozzi, Al-Rfou & Skiena (2014)*.

## Sequence-based approaches

**Definition 5 (Random walk on graph)** *is a sequence of nodes obtained from the random process of node sampling. Usually, probability of choice of node j after node i is proportional to $A_{i,j}$.*

Motivated by drawbacks of the matrix factorization approach, another approach emerged that attempts to preserve local neighborhoods of nodes and their properties based on random walks (*Newman, 2005*; *Pirotte et al., 2007*). More specifically, the main idea is to maximize the probability of observing the neighborhood of a node given its embedding, following the line of Skip-gram model initiated in NLP applications by *Mikolov et al. (2013)*, *Pennington, Socher & Manning (2014)*. An objective of this type can be efficiently optimized with stochastic gradient descent on a single-layer neural network, and hence has lower computational complexity.

**Definition 6 (Skip-gram)** *is method to learn sequence element i representation via maximization of probability of elements in context of i based on representation of i.*

Two prominent examples of models in this class are node2vec (*Grover & Leskovec, 2016*) and DeepWalk (*Perozzi, Al-Rfou & Skiena, 2014*). DeepWalk performs a random walk over a graph and then uses sampled sequences to learn embeddings, using the Skip-gram objective (while having modifications for other NLP based sequence models, such as using Glove from *Brochier, Guille & Velcin (2019)*). Its predecessor LINE (*Tang et al., 2015*) is equivalent to DeepWalk when the size of vertices' contexts is set

to one. Node2vec extends the random walk with biasing parameters of BFS or DFS parameters. Another way of sampling based on diffusion was presented in diff2vec (*Rozemberczki & Sarkar, 2018*). By virtue of sampling being more centered around source nodes, it provides robust embeddings while being less flexible.

Walklets (*Perozzi, Kulkarni & Skiena, 2016*) as a generalization of GraRep (*Cao, Lu & Xu, 2015*) use weighted combination of embeddings of powers of adjacency matrix $A$, $A^2$, ..., $A^k$ to reduce the bias of Deepwalk for low-order proximities, and approximates computing $A^i$ by skipping nodes using short random walks (*Perozzi et al., 2017*).

The focus on the local structure and non-convex optimization requiring the use of stochastic gradient descent and proper initialization limit random walk based methods in capturing the hierarchical structure of a graph. HARP (*Chen et al., 2018b*) proposes a meta-strategy for graph embedding under recursive construction of nodes and edges into condensed graphs with similar global structure. These graphs are used as source initializations for embedding detailed graphs, resulting in the end in proper node and edge embeddings, which can be adopted for improving DeepWalk (*Perozzi, Al-Rfou & Skiena, 2014*), LINE (*Tang et al., 2015*), and Node2vec (*Grover & Leskovec, 2016*) algorithms. It was further generalized for community preserving using Modularity Maximization (*Tang & Liu, 2009*) and supporting large free-scale networks (*Feng et al., 2018*).

Alternatively, Struct2vec (*Ribeiro, Saverese & Figueiredo, 2017*) uses structural similarity without using node or edge attributes but considering graph hierarchy to measure similarity at different scales. *Liu et al. (2020c)* uses rooted substructures of a graph to preserve structural similarity. Diffusion wavelet model to capture structural proximity was suggested in *Donnat et al. (2018)*. Another approach to control hyper-parameters in random-walk methods is Graph Attention (*Abu-El-Haija et al., 2018*) learning multi-scale representation over adjacency matrix powers with the probabilistic approach for learning balancing weights for each power. It was further generalized to its deep learning analog in *Veličković et al. (2017)* and *Liu et al. (2018a)*, see also *Lee et al. (2019)* for details on attention models on graphs.

Extension of Deepwalk to heterogeneous networks was suggested in Metapath2vec (*Dong, Chawla & Swami, 2017*). Modifications of random-walk based methods using node attribute concepts and node proximities were suggested in GenVector (*Yang, Tang & Cohen, 2016b*). With GEMSEC (*Rozemberczki et al., 2018*), the authors extend sequence-based methods with additional K-means objective encouraging clustering structure-preserving in the embedding space and improving overall performance. Discriminative Deep Random Walk (DDRW) (*Li, Zhu & Zhang, 2016*) was suggested for the task of attributed network classification. *Çelikkanat & Malliaros (2019)* generalizes random walk based methods to the case of the exponential family of distributions for sampling strategies.

Sequence-based models, such as node2vec, can obtain high-quality embeddings of structural input graph by sampling node sequences and learning context-consistent embeddings but are not able to capture additional node/edge features while being transductive by their nature.

## Deep learning: graph convolutions

Complex non-regular graphs structure makes graph filtering not as simply defined as on images. In the past decades, researchers have been working on the graph signal processing methods including filtering, wavelets, Fourier transformations using graph spectral domain. The studies on these methods can be found in *Shuman et al. (2013)*, *Ortega et al. (2018a)*.

Advances in deep learning have led to a new field of studies devoted to applying neural networks to graph data (*Scarselli et al., 2009*; *Li et al., 2014a*, *2014b*). Recently, SDNE (*Wang, Cui & Zhu, 2016*) and DNGR (*Cao, Lu & Xu, 2016*) use deep autoencoder to capture non-linearity in graphs and simultaneously apply dimension reduction for constructing graph embedding. SDNE use autoencoder preserving first order proximity and Laplacian Eigenmaps for penalizing long distances for embedding vectors of similar vertices. DGNR uses stacked denoising autoencoders over positive pointwise mutual information matrix obtained from similarity information based on random surfing. Both methods use global information and thus are not appropriate for large networks.

*Kipf & Welling (2016a)* propose Graph Convolutional Layer that offers a further simplified approximation to spectral convolution and achieves better computational efficiency for semi-supervised multi-class node classification is applicable for the other machine learning tasks. A model of several such convolutions is referred to as Graph Convolutional Network (GCN). Improvements over speed and optimization methods of training GCNs were suggested in *Chen, Zhu & Song (2017)*, *Chen, Ma & Xiao (2018)*. Stochastic approaches for network embedding optimization were briefly over-viewed in *Lei, Shi & Niu (2018)*.

Assume the graph $G(V,E)$, adjacency matrix $A$ and feature matrix $X$ of size ($N_{nodes}$, $N_{features}$), where $N_{nodes}$ refers to number of vertices and $N_{features}$ to number of node attributes. Then, GCN can be defined as set of hidden layers $H^i = \sigma(AH^{i-1} W^{i-1})$ where $H^0$ is equal to matrix $X$, $W^i$ is learnable weight matrix. At the next hidden layer, these features are aggregated using the same propagation rule. It means that graph convolutions aggregate feature information of its neighbors based on the adjacency matrix. The idea of graph convolutions using spatial convolutions (operating with adjacency matrix) or spectral graph methods (operating with graph Laplacian) was proposed in *Bruna et al. (2013)*, *Duvenaud et al. (2015)*, *Henaff, Bruna & LeCun (2015)*, *Niepert, Ahmed & Kutzkov (2016)*, *Defferrard, Bresson & Vandergheynst (2016)*, *Levie et al. (2017)*, while extending the GCN idea to recurrent models *Li et al. (2015c)*, *Monti, Bronstein & Bresson (2017)*, mixture models of CNNs *Monti et al. (2017)*; *Fey et al. (2018)*, diffusion convolutions *Atwood & Towsley (2016)*; *Li et al. (2017c)*, and models suitable for dynamic graphs under inductive learning paradigm *Natarajan & Dhillon (2014)*; *Hamilton, Ying & Leskovec (2017a)*. All the methods suggest semi-supervised embedding, however, choosing unique labels for each vertex one may obtain an unsupervised version of network embedding. The GraphSAINT (*Zeng et al., 2019*) provides a solution for scalability problem in training graph neural networks. It compares different topology-based sampling algorithms

(node, edge and random walks) in terms of bias and variance of learned GCN model. It also introduces unbiased estimator for node aggregation.

Another idea is to use deep autoencoders to learn compressed representations that capture the essence of the graph structure. An autoencoder includes two nonlinear functions, an encoder and a decoder, and attempts to minimize reconstruction loss. One such model specifically designed for graphs is GAE, which consists of a GCN encoder (one or two stacked GCN layers in most use cases) that produces embeddings and an inner product decoder that reconstructs the adjacency matrix ($\hat{A} = \sigma(UU^T)$, where σ is non-linearity like sigmoid function and $U$ is embedding matrix of nodes). The weights of the model are trained by backpropagating the reconstruction loss, which is usually Mean Squared Error (MSE).

VGAE (*Kipf & Welling, 2016b*) is a probabilistic counterpart of GAE. It introduces a distribution over latent variables $Z$, with these variables being conditionally independent Gaussians given $A$ and $X$ with means (μ) and diagonal covariances (σ) being parameterized by two GCN encoders (*Kingma & Welling, 2013*). As in the case of images, VGAE just adds KL-divergence term between conditional distribution $q(Z|X,A)$ and unconditional $p(Z) \sim N(0,1)$ to the loss. After node embeddings are reconstructed via random normal distribution sampling, that is, $Z = \mu + \sigma\varepsilon$. Then adjacency matrix is decoded using inner product of achieved vector $Z$ as in simple GAE.

In very recent work, authors of GraphSAGE (*Hamilton, Ying & Leskovec, 2017a*) offer an extension of GCN for inductive unsupervised representation learning and offer to use trainable aggregation functions instead of simple convolutions applied to neighborhoods in GCN. GraphSAGE learns aggregation functions for a different number of hops that are applied to sampled neighborhoods of different depths, which then are used for obtaining node representations from initial node features. PinSage (*Ying et al., 2018a*) extends the previous algorithm with the importance sampling based on random walks. Importance score is calculated simply as visit counts. It provides better scalability and quality. GAT (*Veličković et al., 2017*) use masked self-attention layers for learning weights balancing impact of neighbors on node embedding, and supporting both, inductive and transductive learning settings. In *Liu et al. (2018a)*, authors suggested specific layers controlling the aggregation of the local neighborhood over BFS and DFS sampling, thus generalizing Node2vec (*Grover & Leskovec, 2016*) model to graph neural networks. Similar to GCN, GAT contains several hidden layers $H^i = f(H^{i-1}, A)$, where $H_0$ is a graph node features. In each hidden layer linear transformation of input is firstly calculated with the learnable matrix $W$. The authors replace the adjacency matrix by learnable self-attention in form of a fully-connected layer with activation and further normalization with softmax. Generalization of gated recurrent graph neural networks (*Li et al., 2015c*) was suggested in Message Passing Neural Network (MPNN) (*Gilmer et al., 2017*) providing a differentiable way to combine information from neighbours.

Nowadays, many advanced deep neural network models are adapted to graph data. Graph generative adversarial networks were suggested in *Ding, Tang & Zhang (2018)* and *Yu et al. (2018)*. In *You et al. (2018)*, recurrent graph neural network was suggested for the task of graphs generation. Pooling operators for graphs were used in

*Defferrard, Bresson & Vandergheynst (2016)*, *Ying et al. (2018b)*. *Yuan & Ji (2020)* modernize classic pooling to account graph structure using Conditional Random Fields. Adversarially regularized variational graph autoencoder (ARVGA) was suggested in *Pan et al. (2019)*. *Zhu et al. (2020a)* develop the DGGAN model that jointly learns source and target vectors for the directed graphs employing adversarial techniques. *Liu (2020)* builds Anonymized GCN with adversarial training to be robust to the noise attacks. *Hettige et al. (2020)* propose the RASE model, that applies Gaussian denoising attribute autoencoder for achieving robustness of received embedding, while *Laakom et al. (2020)* catches the uncertainty by learning probability Gaussian distributions over embedding space. *Weng, Zhang & Dou (2020)* employs adversarial training for variational graph autoencoder. *Zhu et al. (2020b)* use node feature smoothing for learn better embeddings. *Jing et al. (2020)* designs variable heat kernel to learn robust representations.

Deep Learning models are now a study of vulnerability to adversarial attacks, in particular, it relates to structural data. The first approaches for detection of node/edge add/remove mechanisms were studied in *Bojcheski & Günnemann (2018)*, *Chen et al. (2018c)*, while other researchers focused on methods for unsupervised (*Sun et al., 2018b*), semi-supervised (*Chen et al., 2018e*) and supervised (*Zügner, Akbarnejad & Günnemann, 2018*) scenarios of graph embedding construction, and application for ML problems. The black-box approach was formulated in *Dai et al. (2018)* and further covered in general overview for the problem of graph data poisoning (*Chen et al., 2019b*) and its applications to social media data (*Zhou et al., 2018*) and knowledge graphs (*Zhang et al., 2019b*). A survey of methods for defense from adversarial attacks on graphs was suggested in *Sun et al. (2018a)*.

The deep learning models propose a new way of approximation for classic graph convolutions and kernels, which allows extracting embeddings faster. A mixture of it with semi-supervised techniques gives the state-of-the-art results in terms of scalability, speed and quality on downstream tasks.

### Hyperbolic (non-Euclidean) embeddings

The Euclidean space is not the best for structures like graphs, because has the low descriptive ability for hierarchical and scale-free structures. So, researchers have considered other space, that can successfully represent it in a comparatively low number of dimensions, saving the basic properties like angles. It allows using classical machine learning methods in down-streamed tasks.

In certain cases, embedding into non-Euclidean spaces may be beneficial for model performance (*Kleinberg, 2007*; *Shavitt & Tankel, 2008*; *Krioukov et al., 2009*). LEs were also used for constructing embedding in hyperbolic space (*Alanis-Lobato, Mier & Andrade-Navarro, 2016*). Deep learning approach was applied for hyperbolic embedding in *Chamberlain, Clough & Deisenroth (2017)*.

There is no exact research on the properties of embedding spaces, but researchers mostly pay attention to preserving low dimensional space, catching graph properties and model quality trade-off.

## SPECIFIC EMBEDDINGS BASED ON NETWORK TYPES

In this section, we show specific embedding models generalizing core methods of network representation to a certain domain of networks and applications based on the network type.

### Attributed networks

Real-world networks are often accompanied with additional features for nodes and edges, such as labels, texts, images. These attributes tend to be correlated for close graph structures and could affect network embedding by adding additional information for the similarity of nodes. The attributes are usually represented by high-dimensional vectors of features (which are sparse for just label attributes). Once the attributes are represented by their embeddings, the task is to incorporate them in network embedding model (under unsupervised or semi-supervised framework).

The authors of TADW (*Yang et al., 2015*) represent DeepWalk model as matrix factorization and incorporate text attributes into factorization framework. PLE (*Ren et al., 2016*) jointly learns the representations of entity types and links together with text features. In *Le & Lauw (2014)*, a generative model for document network embedding was suggested based on topic modeling of documents using Relational Topic Model (RTM) (*Chang & Blei, 2009*) and the relationships between the documents. In *Ganguly et al. (2016)*, authors combine text and network features for co-authorship recommendations.

Augmented Relation Embedding (ARE) (*Lin, Liu & Chen, 2005*) adds content-based features for images using graph-Laplacian spectral embedding modification. In *Geng et al. (2015)*, *Zhang et al. (2015, 2017)*), authors suggested to embed images, textual and network information for modeling user-image interaction.

In addition to structural similarity, in certain cases feature similarity may be also important. Two-layered network embedding for node-to-node and text-to-text similarities was suggested in *Sun et al. (2016)*. In *Zhang et al. (2016b)*, the authors proposed the HSCA model, embedding homophily, network topological structure and node features simultaneously. In DeepBrowse (*Chen, Anantharam & Skiena, 2017*), the authors suggested using DeepWalk-based node similarity together with priority ranking for recommender system based on an interaction graph. Label preserving attribute node embedding was suggested in Tri-party Deep Network Representation (*Pan et al., 2016*). Modifications of random-walk based methods using node attribute concepts and node proximities were suggested in GenVector (*Yang, Tang & Cohen, 2016b*).

Label attributes are also an important part for such problems as classification of nodes and edges, or community information (assigning each node a community label). Community preserving network embeddings were suggested in *Shaw & Jebara (2009)*, *Wang et al. (2017d)* and *Rozemberczki et al. (2018)*. Incorporating group information was presented in GENE model (*Chen, Zhang & Huang, 2016b*) under a supervised framework.

Semi-supervised frameworks for learning network embedding under loss constraints for labeled data were suggested in Planetoid (*Yang, Cohen & Salakhutdinov, 2016a*) Max-margin Deep Walk (*Tu et al., 2016*) and LANE (*Huang, Li & Hu, 2017*).

## Heterogeneous networks

A heterogeneous network presents a different concept of graph representation, in which nodes and edges may have different types (or even multiple edges).

The heterogeneous network embeddings either learn embeddings in the same vector space (*Li, Ritter & Jurafsky, 2015*; *Zhao, Liu & Sun, 2015*), or construct the embeddings separately for each modality and then aggregate them into one space, such as HNE model (*Chang et al., 2015*) and *Tang & Liu (2011)*, or even aggregate over multiple network layers (*Xu et al., 2017*) or different relation features (*Huang, Li & Hu, 2017*).

Random-walk based approach for different node types based on DeepWalk was presented in Metapath2vec (*Dong, Chawla & Swami, 2017*). Similar approaches based on meta-path random walks for graph embedding were suggested in *Huang & Mamoulis (2017)*, *Chen & Sun (2017)*. *Jacob, Denoyer & Gallinari (2014)* use heterogeneous network embedding for node classification across different node types. A similar problem was posed for author identification on double-blind review scenario (*Chen & Sun, 2017*). Study by *Jiang et al. (2020)* provides a framework for efficient task-oriented skip-gram based embeddings. *Hu, Fang & Shi (2019)* utilizes the generative adversarial networks, which learn node distributions for efficient negative sampling. *Shi et al. (2020)* proposes a method for automatic meta-path construction.

*Cao et al. (2020)* use the graph attention mechanism for heterogeneous graph embedding task. MAGNN architecture (*Fu et al., 2020*) extends simple attention mechanism with several levels: node attributes, inter meta-path information and intra meta-path semantic information. DyHAN (*Yang et al., 2020*) presents the model for dynamic heterogeneous graphs with hierarchical attention. Another way to use the attention mechanism in dynamic heterogeneous networks is the *Li et al. (2020b)*. It employs three types of attention: structural, semantic and temporal.

Heterogeneous graph embeddings are widely used in real-world applications. *Hong et al. (2020)* estimates the arrival time for transportation networks, *Ragesh et al. (2020)* use it in text classification. *Chen & Zhang (2020)*, *Li et al. (2020a)* utilizes HIN embedding for multi-modal data fusion task. *Zhang et al. (2020a)* preserves the relationships in HIN. A survey on heterogeneous networks can be found in *Wang et al. (2020a)*.

## Signed networks

In a signed network, each edge is associated with its weight, taking values from the set $\{1, -1\}$, which usually represents belief or opinion sentiment for different relations types. These networks are specifically considered apart from Heterogeneous networks as important objects for social network analysis, although they are still just a specific type of such networks. One of the tasks on such networks is predicting links and their signs (*Liu et al., 2015a*).

SiNE (*Wang et al., 2017c*) is a DNN model aiming at close relationships with friends (positive weight) rather than with foes (negative weight). For highly positive social networks a virtual node with negative relation is proposed to use in the model, which uses pairwise similarities optimization under constraint mentioned above. In *Yuan, Wu & Xiang (2017)*, the authors propose a local neighborhood aggregation model SNE for each

type of positive and negative relations. *Kim et al. (2018)* propose random-walks based model SIDE for signed directed networks. Also, they provide socio-psychological interpretation for each term in the loss function. SIGnet (*Islam, Prakash & Ramakrishnan, 2018*) develops new target node sampling for more efficient learning. In oppose to previous works, *Lu et al. (2019)* provides signed network embedding powered by Status Theory (*Leskovec, Huttenlocher & Kleinberg, 2010*). It natively works with directed networks by preserving node ranking except direct node similarity.

## Multi-layer networks

Multi-layer networks are used to model complex systems with different levels of interaction between nodes, for example, whole Airline network with different carriers. Each layer in such networks corresponds to different types of relationships.

*Liu et al. (2017a)* compare three aggregation methods for single-layer network embedding models: merging of different layers in one network, single-layer vectors concatenation and between-layer random walks. The best results show the last method named layer co-analysis because it allows learning between-layer interactions. In *Xu et al. (2017)* authors provide an example of coupling into joint space two separately learned heterogeneous networks embeddings. IONE (*Liu et al., 2016*) preserves users similarity based on their followers and followees for several social networks. A hierarchy-aware unsupervised node feature learning approach for multi-layer networks was proposed in *Zitnik & Leskovec (2017)*. In *Li et al. (2018)* authors develop the single optimization framework for both within-layer and between-layer communication. It exploits spectral embedding and the block model.

## Temporal networks

A lot of real-world networks are evolving over-time. Most of the described above methods concentrate on the static embeddings, so it works poorly in the temporal scenario.

*Haddad et al. (2019)* propose the adaptation of Node2vec model to the dynamic case. Authors also introduce the task-specific temporal embeddings. *Rossi et al. (2020a)* provide the generic framework named Temporal Graph networks for deep learning on dynamic graphs. *Fathy & Li (2020)* apply the graph attention to the temporal networks. *Zhong, Qiu & Shi (2020)* develop the model for efficient community mining. *Rokka Chhetri & Al Faruque (2020)* present the model for dynamic physics graphs. CTGCN model (*Liu et al., 2020a*) generalizes graph convolution networks with feature transformation and aggregation. It builds the hierarchical representation of the graph with K-cores and applies GCN to it. *Goyal, Chhetri & Canedo (2020)* use the recurrent neural networks to catch the dynamics. There is one more specific graph type: temporal interaction networks, such as user-item interactions in the recommender systems. *Zhang et al. (2020b)* creates the embedding approach for such graph utilizing coupled memory networks.

Nowadays, methods based on smart neighborhood aggregation, such as limiting random walks over clusters *Chiang et al. (2019)* and precomputing diffusion-based neighborhoods for one-layer GCN *Rossi et al. (2020b)* show great performance over existing approaches, thus combining advances in deep learning and neighborhood sampling methodology.

## Large graphs

We have already mentioned that random walks and graph neural networks were proposed as the approximations for the different classic matrix factorization techniques. So in this section, we will discuss approaches to scale up GNN training.

The basic idea implemented in different papers is a sampling. GraphSAGE (*Hamilton, Ying & Leskovec, 2017a*) learns trainable aggregations for sampled node neighbourhood. This approach was further improved with fixed-length random walk based importance sampling of the neighborhood in *Ying et al. (2018a)*. GraphSAGE also provides the idea of minibatch training for GNNs. A similar idea was proposed in the *Chen, Ma & Xiao (2018)*. *Salha, Hennequin & Vazirgiannis (2020)* propose to use linear aggregation over direct neighbors to simplify computations. The GraphSAINT (*Zeng et al., 2019*) compares different topology-based sampling algorithms (node, edge and random walks) in terms of bias and variance of learned GCN model. It also introduces unbiased estimator for aggregation of node and normalizes propagation by this value, that solves the scalability problem.

*Nie, Zhu & Li (2020)* is based on the idea of Locality Preserving Projection. It works with anchor-based proximity matrices and calculates these anchors via Balanced and Hierarchical K-means. Such an approach allow to reduce complexity from $n^2d$ to $ndm$ where $n$ is a number of samples, $d$ is embedding dimension and $m$ is a number of anchors. *Akyildiz, Aljundi & Kaya (2020)* extends the VERSE (*Tsitsulin et al., 2018*) with graph partitioning and coarsening to provide fast embedding computation on the GPU. *Atahan Akyildiz, Alabsi Aljundi & Kaya (2020)* analyzes effects of graph coarsening on different embeddings in comparison to GOSH. Another distributed training framework was presented in *Zheng et al. (2020a)*. It also provides efficient graph partitioning schemes for reducing between-machine communication. *Gallicchio & Micheli (2020)* keeps the graph embedding as the dynamical systems and study the embedding stability issue. Authors found that stable initialization allows to left weights untrained in deep sparse networks. *Lu & Chang (2020)* use softmax clustering for modularity maximization. They show that such a method is a linear approximation for main eigenvectors.

# APPLICATION OF GRAPH EMBEDDINGS TO MACHINE LEARNING PROBLEMS

Here, we aim to overview core machine learning problems involving structural data. We start with problems related to small graph motifs such as nodes and edges, while further going to the problems connected to subgraphs and graphs as a whole.

## Node classification

**Definition 7 (Node classification)** *For a given graph G(V, E) with known labels for some of nodes from V, node classification is the task of predicting missing labels for existing or newly added nodes.*

Node classification deals with assigning class labels to nodes based on labeled nodes data (*Zhu et al., 2007*; *Bhagat, Cormode & Muthukrishnan, 2011*). The structural information is used in a context that "similar" nodes should have the same/similar labels. The original

framework uses label propagation based on random walks statistics (*Xiaojin & Zoubin, 2002*; *Azran, 2007*; *Baluja et al., 2008*). In an unsupervised framework, each node is embedded in a low-dimensional space following by training a classifier on the set of labeled node embedding vectors (*Lu & Getoor, 2003*; *Bhagat, Cormode & Rozenbaum, 2009*). Authors use such machine learning models as logistic regression (*Perozzi, Al-Rfou & Skiena, 2014*; *Pimentel, Veloso & Ziviani, 2017*), SVM (*Wang, Cui & Zhu, 2016*; *Wang et al., 2017d*), kNN (*Le & Lauw, 2014*; *Wilson et al., 2014*), random forest and xgboost (*Makarov et al., 2018*; *Makarov et al., 2019c*); the choice is usually made based on the size of training data, interpretability of features and embedding dimension.

In semi-supervised framework, node embeddings are learned via loss function containing regularization for labeled data predictions, penalizing "similar" nodes to have different labels (*Li, Zhu & Zhang, 2016*; *Yang, Cohen & Salakhutdinov, 2016a*; *Tu et al., 2016*; *Kipf & Welling, 2016a*; *Monti et al., 2017*). *Zhang, Zhou & Li (2020)* proposes hierarchical GCN and pseudo-labeling technique for learning in scarce of annotated data. *Liu et al. (2020b)* proposes a sampling strategy and model compression for handling sparsity of labels. *Chen et al. (2020)* employs contrastive learning techniques to achieve semi-supervised parametrized fusion of graph topology and content information. *Zhu et al. (2020c)* also use metric learning approach but applies it to corrupted graph substructures. *Nozza, Fersini & Messina (2020)* use two-phase optimization for attributed graph embedding. *Shi, Tang & Zhu (2020)* aligns topology of attribute content network to the corresponding graph to simultaneously learn good embeddings. *Wang et al. (2020b)* propose two models for the imbalanced scenarios. A survey on classic techniques for node classification can be found in *Bhagat, Cormode & Muthukrishnan (2011)*.

## Link prediction

**Definition 8 (Link prediction problem (LPP))** *is a task of completing missing edges in noisy graphs or predicting new edges in temporal network structures. Formally, LPP for given graph G(V, E) with adjacency matrix A is a task of learning such function f that reconstruct or predict next adjacency matrix A based on different graph features such as metrics (e.g., Jaccard, Adamic-Adar), graph embeddings.*

Network science approach to the problem of predicting collaborations results in the link prediction (LP) problem (*Liben-Nowell & Kleinberg, 2007*) for temporal networks and missing edges reconstruction in noisy network data. Basically, it is a method to apply standard machine learning framework for graph data considering feature space consisting of pairs of nodes and their features.

One of the interesting research questions is in the way of constructing edge embedding in a non-direct combination of node embeddings, as it was suggested in component-wise embeddings (*Grover & Leskovec, 2016*) or bi-linear combination of compressed node embeddings suggested in *Abu-El-Haija, Perozzi & Al-Rfou (2017)*. Certain practical applications for drug combinations was suggested in *Zitnik, Agrawal & Leskovec (2018)*.

HARP *Chen et al. (2018b)* incorporates several hierarchical layers while transmitting information from edge embedding to node embedding. Other systems of directly

incorporating edge features and labels were suggested in CANE (*Tu et al., 2017*) and LANE (*Huang, Li & Hu, 2017*). Models of joint node and edge structure learning were proposed in Dual-Primal GCN (*Monti et al., 2018*) and ELAINE (*Goyal et al., 2018*). A model for embedding event graphs in which event is described by several edges was presented in HEBE (*Gui et al., 2016*). *Wu et al. (2020)* presents random walk with restart index. *Phuc, Yamada & Kashima (2020)* embeds several graphs with similar structural properties to boost link prediction accuracy. *Keser et al. (2020)* employs skip-connections in VGAE.

Link prediction models are applied in web linking (*Adafre & de Rijke, 2005*), social dating services (*Backstrom & Leskovec, 2011*) and paper recommender system for digital libraries (*He et al., 2010*). The reader can found an up-to-date survey in *Srinivas & Mitra (2016)*.

LPP was specifically formulated in *Liben-Nowell & Kleinberg (2007)* based on nodes pairwise similarity measures. Approaches for link prediction include similarity based methods (*Adamic & Adar (2003)*), maximum likelihood models (*Clauset, Moore & Newman, 2008*), and probabilistic models (*Getoor & Taskar, 2007*; *Heckerman, Meek & Koller, 2007*). In *Tang & Liu (2012)*, authors are suggesting unsupervised approach for LP problem. *Gao, Denoyer & Gallinari (2011)*, *Gao et al. (2015)* suggested temporal link prediction based on matrix factorization technique and noise reduction in large networks. Attribute-based link formation in social networks was studied in *McPherson, Smith-Lovin & Cook (2001)*, *Robins et al. (2007)*, while deep learning approaches were presented in *Liu et al. (2013)*, *Zhai & Zhang (2015)* and *Berg, Kipf & Welling (2017)*. Heterogeneous graph link prediction for predicting links of certain semantic type was suggested in *Liu et al. (2017b, 2018b)*. An evaluation of link prediction models based on graph embeddings for biological data was presented in *Crichton et al. (2018)*.

Two surveys on link prediction methods describing core approaches for feature engineering, that is, Bayesian approach and dimensionality reduction were presented in *Hasan & Zaki (2011)* and *Lü & Zhou (2011)*. Survey on link prediction was published in *Wang et al. (2015)*.

## Node clustering

**Definition 9 (Node clustering or community detection or graph partitioning)** *is the task of the partitioning of a graph $G(V, E)$ into several subgraphs $G_i(V_i, E_i)$ with a dense connection within groups and sparse connection between clusters.*

Node clustering (also known as community detection in social network analysis) aims to find such a grouping (labelling) of nodes so that nodes in the same group are closer to each other rather than to the nodes from outside of the group (*Malliaros & Vazirgiannis, 2013*). No labels are provided on initial step due to unsupervised type of the problem. Methods use attribute (*Zhou, Cheng & Yu, 2009*) or structural information. The latter methods of graph clustering are usually based on either community detection (*Newman & Girvan, 2004*; *Fortunato, 2010*) or structural equivalence (*Xu et al., 2007*). In community detection (*Shi & Malik, 2000*; *Ding et al., 2001*), the cluster is defined as

dense subgraph with a high number of edges inside subgraph, and a low number of edges between subgraph and the rest of a graph.

The general idea is to use node embeddings as a compressed representation of sparse graph adjacency matrix and then apply standard clustering algorithms, such as K-means or DBScan, for vectorized data (*White & Smyth, 2005*; *Tian et al., 2014*; *Cao, Lu & Xu, 2015*; *Chen et al., 2015b*; *Cao, Lu & Xu, 2016*; *Nie, Zhu & Li, 2017*). Going further, joint optimization of clustering and node embedding was suggested in *Tang, Nie & Jain (2016)*, *Wei et al. (2017)*. Efficient iterative community aware network embedding was proposed in *Wang et al. (2017d)* and several others (*Zheng et al., 2016*; *Cavallari et al., 2017*).

*Teng & Liu (2020)* propose multi-objective evolutionary algorithm for community detection. *Zhang, Shang & Jiao (2020)* use multi-objective matrix factorization over several shortest path graphs and utilizes (MOEA) to find community structure. *Salim, Shiju & Sumitra (2020)* train the embeddings on different views for preserving many properties of a given network. *Quiring & Vassilevski (2020)* employs hierarchical coarsening of the graph to better extract clusters.

## Subgraph (and graph) embedding

While studying network embedding, one may think of a way to aggregate or generalize low-level node feature representation to the whole network representation, thus stating the problem of embedding the whole graph (*Song, 2018*). Such vector is required for the graph-level tasks like graph classification, similarity and clustering. It considers the whole network as one structural unit in the training dataset.

The task is relevant to chemistry or biology domains (*Nikolentzos, Meladianos & Vazirgiannis, 2017*; *Zhang et al., 2016a*; *Duvenaud et al., 2015*; *Dai, Dai & Song, 2016*; *Niepert, Ahmed & Kutzkov, 2016*; *Kearnes et al., 2016*). They can also be applied for graph reasoning (*Li et al., 2015c*) or computer vision tasks (*Bruna et al., 2013*).

In *Duvenaud et al. (2015)*, the sum based approach over network embedding was suggested. Following by it, in *Dai, Dai & Song (2016)*, authors proposed neural network aggregation for constructing network embedding which is an argument for summing over subgraph nodes. Improvement of these methods was later suggested in *Bronstein et al. (2017)* based on approximations of spectral graph decompositions. Ordered-based (*Niepert, Ahmed & Kutzkov, 2016*) and fuzzy-based (*Kearnes et al., 2016*) approaches based on aggregating features from convolutional approaches further improved subgraph embedding models. *Sun, Hoffmann & Tang (2019)* maximize the mutual information between embedding and different graph substructures.

The general approach of *Gilmer et al. (2017)* as well as other convolutional approaches can be generalized by pooling-aggregation models or, as was suggested in *Scarselli et al. (2009)*, by adding super-node for whole graph embedding. The attention mechanism was applied to the graph classification task (*Lee, Rossi & Kong, 2018*).

**Definition 10 (Line (dual) graph)** *For a **graph** G = (V, E) defined as a set of vertices V and a set of edges $E \subseteq V \times V$ without loops and multi-edges we denote by $G^* = (V^*, E^*)$ a **dual (Line) graph** the nodes of which are the edges of G and edges are nodes, in the*

*sense that two adjacent nodes are connected by an edge if corresponding edges have a common node incident to them.*

In graph-level tasks, specific network properties play a major role. So vectors reconstructing sophisticated similarity metrics closely related to the problem of graph isomorphism was studied in several works (*Shervashidze et al., 2011*; *Niepert, Ahmed & Kutzkov, 2016*; *Mousavi et al., 2017*; *Yanardag & Vishwanathan, 2015*; *Narayanan et al., 2016*). GL2VEC (*Chen & Koga, 2019*) extends *Narayanan et al. (2016)* model with edge features by utilizing the line graph. The works on matching node embedding and graph kernels were suggested in *Johansson & Dubhashi (2015)*, *Nikolentzos, Meladianos & Vazirgiannis (2017)*. In *Donnat & Holmes (2018)* authors analyze graph-based distance methods for a temporal graph of bio-medical surveys.

Hierarchical clustering and fusion of different network representations were overviewed in *Yang & Wang (2018)*. Usually, this kind tasks require fusion of different similarity representations of a network as different graphs (*Serra, Greco & Tagliaferri, 2015*; *Xue et al., 2015*), preserving graph structure (*Hou et al., 2017*) or simultaneously performing semi-supervised classification and clustering with adaptive kNN model (*Nie, Cai & Li, 2017*). Different domain network clustering was suggested in *Cheng et al. (2013)* and improved in the following works suggesting fusion of different not-synchronized networks with different structures (*Ni et al., 2016*), cross-domain associations (*Liu et al., 2015b*) or multi-view spectral clustering (*Li et al., 2015b*). *Khasahmadi et al. (2020)* propose a memory layer for graphs, that can efficiently learn graph hierarchical representations. *Tsitsulin, Munkhoeva & Perozzi (2020)* propose an algorithm for efficient calculation of spectral distances for large graphs. *Kolouri et al. (2020)* suggest the embedding preserving Wasserstein distance with linear complexity. *Qin et al. (2020)* presents one more graph pooling technique that uniformly aggregates neighborhood. *Baldini, Martino & Rizzi (2020)* embeds maximal cliques to preserve structural similarities between graphs. *Yan & Wang (2020)* states the problem of transfer learning suggesting the framework for graph alignment and further adaptation learning for GNNs.

## Network visualization

**Definition 11 (Graph visualization)** *is a way to map a graph to a low (2D, 3D) dimensional space.*

All nodes are either directly embedded as 2D vectors (*Le & Lauw, 2014*; *Wang, Cui & Zhu, 2016*; *Cao, Lu & Xu, 2016*; *Tu et al., 2016*; *Niepert, Ahmed & Kutzkov, 2016*; *Pan et al., 2016*) or first embedded to certain dimension, and then compressed via PCA (*Herman, Melançon & Marshall, 2000*) or t-SNE (*Maaten & Hinton, 2008*) (or other dimension reduction frameworks, see for, for example, *Tenenbaum, De Silva & Langford, 2000*, *De Oliveira & Levkowitz, 2003*) in order to plot in 2D space. If there are labels or communities representative for network dataset, the nodes are usually visualized with different colors for each label in order to verify whether similar nodes are embedded closer to each other. Such models, as *Perozzi, Al-Rfou & Skiena (2014)*, *Grover & Leskovec (2016)*, *Tang et al. (2015)*, *Ou et al. (2016)*, *Wang, Cui & Zhu (2016)* demonstrated proper

performance on the task of network visualization for unsupervised graph embedding models. Evaluation of graph embeddings for large structural data visualization can be found in *Tang et al. (2016a)*. Graph visualization techniques beyond planar mappings can be found in *Didimo, Liotta & Montecchiani (2019)*.

### Network compression

**Definition 12 (Network compression, simplification or sparsification)** *is a task of reducing the number of nodes and edges in a graph, for further efficient application of graph algorithms.*

The concept of network compression was first introduced in *As Feder & Motwani (1991)* under the idea of reducing the number of stored graph edges while achieving a faster performance of certain algorithms on graphs. The compression was made by grouping nodes and edges into partitions of bipartite cliques and then replacing these cliques with trees. Similar ideas of dividing the graph into groups of nodes and edges and encoding them were proposed in several studies (*Pardalos & Xue, 1994*; *Tian, Hankins & Patel, 2008*; *Toivonen et al., 2011*). Minimum Description Length (MDL) (*Rissanen, 1978*) was used in *Navlakha, Rastogi & Shrivastava (2008)* to construct graph summary adjusted with edge correction algorithm.

Graph embeddings support compact graph representation, reducing memory storage from $O(|V| \times |V|)$ to $O(d \times |V|)$, where embedding dimension $d \ll n$ below 200 was shown to be enough for qualitative network reconstruction for second-order preserving proximity models (e.g., link prediction), such as *Ou et al. (2016)* and *Wang, Cui & Zhu (2016)*. They also suit for various graph optimization task providing useful tools for constructing graph-based heuristics (*Khalil et al., 2017*).

## APPLICATIONS TO REAL-WORLD PROBLEMS

In this section, we are interested in how graph embeddings appear in many other computer science fields, in which graphs are not directly expressed in the data, but relations between the objects can be efficiently described by graphs, and so, graph embeddings.

### Computer vision

Image classification can be solved with classic CNN models considering the images as a grid-like structure. Recently, graph convolutional network models can take into account different neighboring relations, thus going beyond the nearest pixels as the only features for convolutions. Especially interesting results were obtained for 3D shape reconstruction (*Monti et al., 2017*) and video action recognition.

There are four main ideas of using graph neural networks for computer vision tasks: working with the interaction of objects on video and images, feature similarity graph, label graph, that is, images with the same label are connected, and internal graph-structured image data.

One of the main problems with CNN is that they should be deep enough to account interaction information between object, so *Chen et al. (2019c)* propose GloRe unit that applies GCNs over interaction data. It helps to efficiently solve relational reasoning task. In

*Wang et al. (2018)* relation graph of image objects was built for localizing object instance from natural language expression. Graph representation is also useful for representing in-label object interaction like in metric learning. It successfully applied to face clustering task (*Yang et al., 2019*; *Wang et al., 2019b*). Also such graph was exploited by *Kim et al. (2019)* for few-shot learning classification. Graph Convolutions are widely used in skeleton-based action recognition. It applies different graph network models to human skeleton graph (*Shi et al., 2019*; *Si et al., 2019*; *Li et al., 2019a*). GNNs are used for video tracking and classification tasks (*Zhang et al., 2018a*; *Gao, Zhang & Xu, 2019*; *Zhong et al., 2019*).

## Natural language processing

NLP is highly correlated to graph tasks. Here similar sequential methods are used, while data have hierarchical structure from different views.

In *Marcheggiani & Titov (2017)*, authors assign semantic roles by encoding sentences with the graph convolutional network. In *Marcheggiani, Bastings & Titov (2018)*, *Zhao et al. (2019)* graph convolutional network models were applied for machine translation. *Sevgili, Panchenko & Biemann (2019)* use the Wikipedia link graph between entities to improve the quality of entity disambiguation task on unstructured text data. Graph models are widely used in NLP to extract syntactic and semantic information (*Luo et al., 2019*; *Vashishth et al., 2019*; *Veyseh, Nguyen & Dou, 2019*). The main approach is to extract the dependency graph and learn node (word) embeddings using GCN. Another approach is to examine each sentence as a complete graph with adjacency weighted by attention.

Graph neural networks also help in sequence tagging task, because it natively exploits information about the connection between different entities. *Zhu et al. (2019)* propose the Generated Parameters GNN for the Relation extraction task. It also builds a complete graph of entities in the sentence via encoding of the sentence with any sequence model. After that, GNN is applied to solve the node classification task. A prominent application of GNNs is to encode dependency tree information. Such an approach is exploited by *Guo, Zhang & Lu (2019)*, they apply Graph Attention Models. *Sahu et al. (2019)* also use dependency graph for relation extraction tasks, but their model accounts for inter-sentence dependencies.

Question answering, comment generation and dialog systems are highly dependent on domain knowledge-base. Such knowledge-base usually can be depicted as knowledge graphs. *Banerjee & Khapra (2019)*, *Kim, Kim & Kwak (2018)* applies GNN to encode knowledge and account to it in these tasks. *Li et al. (2019b)* also use graph models based on news interaction graphs.

The transformer-based language models (*Vaswani et al., 2017*) works in a similar way to graph attention networks. It models a sentence as a complete graph and calculates new word representation weighting previous vectors with self-attention. The BERT model (*Devlin et al., 2018*) is a special case of transformer-based models. It learns the vector by predicting masked words. Such tasks can be formulated as link prediction between context and masked words.

## Knowledge graph completion

Knowledge graph embedding aims to learn vectors for entities and multi-dimensional vectors for entity relations. Knowledge graph completion solves link prediction between entities in knowledge graphs thus predicting ordered triples of entity-relation-entity (*Lin et al., 2015*). Knowledge graph (KG) embedding presents a knowledge base as a collection of triples "head-relation-tail" and consider them training samples. Structured Embedding (*Bordes et al., 2011*) learns two separate entity-relation representations for head and tail, while Semantic Matching Energy (*Bordes et al., 2012*), Latent Factor Model (*Jenatton et al., 2012*) and Neural Tensor Network (*Socher et al., 2013*) embed entities and relations, and use models to capture correlations between them.

A survey on KG embeddings *Wang et al. (2017a)* considers translation-based models, such as TransE (*Bordes et al., 2013*), TransH (*Wang et al., 2014*), TransM (*Fan et al., 2014*), TransR/CTransR (*Lin et al., 2015*), TransC (*Lv et al., 2018*), TransD (*Ji et al., 2015*), TranSparse (*Ji et al., 2016*), KG2E (*He et al., 2015*), and semantic matching models, based on RESCAL (*Nickel, Tresp & Kriegel, 2011*) tensor factorization framework, such as DistMult (*Yang et al., 2014*), HolE (*Nickel, Rosasco & Poggio, 2015*) and ComplEx (*Trouillon et al., 2017*) with comparison paper for the latter two in *Trouillon & Nickel (2017)*.

Question answering via knowledge graph embeddings was suggested in *Huang et al. (2019)*. Weighted attention for supporting triple in KG link prediction problem was presented in *Mai, Janowicz & Yan (2018)*.

## Data mining

*Ye et al. (2019)* proposed method that models relations between different entities in Android logs (API, apps, device, signature, affiliation) using a hierarchical graph. Then they classify nodes of such graphs for real-time malware classification. Graph neural networks are widely used to utilize the social network information. *Wu et al. (2019a)*, *Song et al. (2019)*, *Chen et al. (2019a)* use such models to account for social effects in recommender systems. *Zhang, Ren & Urtasun (2019)* propose Graph HyperNetworks for neural architecture search. It learns topology of architecture and infers weights for it.

## Recommender systems

The basic approach for recommending top K nodes of interest for a given node is usually based on certain similarity metric (*Pirotte et al., 2007*; *Zhao et al., 2013*; *Gui et al., 2016*; *Zhou et al., 2017*; *Ou et al., 2016*). There are various situations in which one need to provide node recommender system *Zhang & Wang (2016)*, in particular, for items to customers via APP model (*Zhou et al., 2017*), documents matching a given query (*Xiong, Power & Callan, 2017*), community-based question answering (*Zhao et al., 2016*; *Fang et al., 2016*), music recommendations via user preference embedding for query answering (*Chen et al., 2015a*, *2016a*), location recommendations (*Xie et al., 2016*), and many other real-world scenarios.

Matrix completion approach based on graph embeddings was provided in *Monti, Bronstein & Bresson (2017)*. Large scale recommender system was presented in *Ying et al. (2018a)*. Explainable recommendations were studied in *Zhang & Chen (2018)*.

In *Zhang, Wang & Zhang (2019d)* authors represents product search as a graph of co-clicked answers. They mix network embedding, term item vectors and term query vector using MLP to predict the probability of click on the item in certain query. This score is used to rank products.

STAR-GCN (*Zhang et al., 2019c*) is used over user-item interaction graph to learn user and item vectors. This approach is also suitable for inductive learning only using several interactions of users and items. This helps to solve the cold-start problem in recommender systems. *Shang et al. (2019)* use graphs for encoding hierarchical structure of health diseases. Next, achieved embeddings are integrated into BERT model for visit-based user recommendation.

The classical specific case of using network science in recommendations is the link prediction in collaborator networks (*Chen, Li & Huang, 2005*; *Liu & Kou, 2007*; *Li & Chen, 2009*; *Cho & Yu, 2018*). *Kong et al. (2018)* developed a scientific paper recommender system based on citation networks, which uses text information embeddings to find papers of similar research interest and structural network embedding. The combined embedding model was then applied for constructing article vector representations. A combination of network and knowledge graphs was proposed in *Yang, Tang & Cohen (2016b)*. In *Makarov et al. (2019a*, *2019b*, *2019c)* authors show that two-level architecture can improve the recommendation results. Firstly it predicts the collaboration itself and further estimates its quantity/quality. A survey on co-authorship and citation recommender systems may be found in *Ortega et al. (2018b)*.

## Biomedical data science

The large variety of data in biomedicine can be represented as networks. *Le, Yapp & Yeh (2019)* applies embedding techniques to electron transport chains. *Do, Le & Le (2020)* utilizes it for detection of specific proteins. *Lin et al. (2020b)* exploits the dynamic graph embedding for detecting changes in functional connectivity in the brain network.

Computational drug design is an attractive direction because it reduces the costs of development of new drugs. The prominent field is drug repositioning. It usually works with networks of drug interaction with other entities: target, disease, gene or another drug. The main idea of such task is to predict possible relations between drug and other entities *Su et al. (2020)*. For example, drug-disease interaction networks can predict the possible treatment of new disease with existing drugs. So, it is a similar statement to the link prediction problem. *Yamanishi et al. (2008)*, *Cobanoglu et al. (2013)*, *Ezzat et al. (2017)* find drug-target pairs via proximity over matrix factorization based embeddings. *Zheng et al. (2013)*, *Yamanishi et al. (2014)*, *Ezzat et al. (2016)* try to add external data to the drug-interaction network embeddings. *Luo et al. (2017)*, *Zong et al. (2017)*, *Alshahrani et al. (2017)* build heterogeneous networks of different drug-related interaction and apply network embedding methods to it. *Wang et al. (2019a)* embeds heterogeneous gene graph to predict drug response.

Another important field in medicine design is the adverse drug reaction (ADR) analysis. Some articles (*Zitnik & Zupan, 2016*; *Zitnik, Agrawal & Leskovec, 2018*) focus on similar drug–drug and drug–target interaction prediction. *Wang (2017)*, *Abdelaziz et al. (2017)*

use the knowledge graph based on biomedical texts. *Stanovsky, Gruhl & Mendes (2017)* also works with KG embedding, but over ADRs mentions in social media.

Network science is also applied to the molecule structure. *Li et al. (2017b)* proposes a prediction of pathogenic human genes using network embedding. Network embedding is very popular method in protein–protein interaction assessment and function prediction (*Kulmanov, Khan & Hoehndorf, 2018*; *Su et al., 2020*; *Wang, Qu & Peng, 2017b*). *Shen et al. (2017)* and *Li et al. (2017a)* applies to miRNA-disease interaction network to associate genes with complex diseases. The detailed survey of biomedical network embedding applications is presented by *Su et al. (2020)*.

### Reinforcement learning

Reinforcement learning (RL) is a popular approach to solve combinatorial optimization problems. *Zheng, Wang & Song (2020)* provides the open-sourced environment for graph optimization problems using reinforcement learning and graph embeddings. *Hayashi & Ohsaki (2020)* use RL for a similar task, such as binary topology optimization of trusses. It utilizes graph convolution networks for feature extraction and further usage in RL optimization. A similar concept was used in *Yan et al. (2020)* to solve automatic embedding problem using actor-critic models for optimization and graph embeddings for representation learning.

*Waradpande, Kudenko & Khosla (2020)* suggests encoding states in Markov's decision process with graph embedding models. *Lin, Ghaddar & Nathwani (2020a)* follows this idea and utilizes GNN for parametrization of the stochastic policy in electric vehicle routing problem. *Zhou et al. (2020)* solves the interactive recommender system problem enhancing it with knowledge graphs. It describes states using GCN over knowledge graph.

## OPEN PROBLEMS

Here we mention the most interesting open problems in graph representation theory, which are far from good results applicable for any given scenarios.

Many real-world graphs are dynamic: nodes and edges can appear and vanish over time. Despite a large number of recent papers, this field is far from benchmark well-performing models as of now. One of the approaches for it is inductive learning, which is strongly correlated with graph dynamics problem. Inductive methods allow finding embedding for newly added nodes without refitting the whole model. It is important in real-world applications and partially solve the scalability issue.

Edge attributes aware network embedding is poorly studied field. There is a low number of models. Such models usually depend on a Line graph, which has a dramatically larger number of nodes. So such models have a problem with scalability. Edge attributes are important in such tasks as context-aware recommender systems or transportation networks optimization.

They are an only little number of works about subgraph embedding. Such models should represent complex structures like triangles or hierarchy. The application of non-euclidean spaces to the embedding task is a promising method solving this issue, but also poorly studied.

Recent advances in the distributed and batch training for graph neural networks looks promising. However, most of the methods are not theoretically grounded, so it could be hard to understand the issues of poor quality of results. Only *Zeng et al. (2019)* provides some bias-variance analysis of node and edge sampling approaches. However, *Akyildiz, Aljundi & Kaya (2020)* provides a much faster and powerful method for large scale embedding.

Another field that is currently under the control of many papers is the heterogeneous graph embedding. Such graphs are very common in real-world scenarios. The graph attention-based methods look promising in that field. It allows us to catch different aggregation levels like in *Fu et al. (2020)* and *Li et al. (2020b)*.

As can be seen from our survey, most embedding models catch specific graph attributes and there is no general model, thus, raising a problem of selection and recommendation of different models for specific use-cases.

It is also an interesting point to develop meta-strategies for embedding mixture, that will preserve different graph properties. Such meta-models could solve the problem of knowledge generalization and reduce costs for deploy of application.

As in the other fields like NLP and CV, graph neural networks are poorly interpretable, apart from an initial study in *Ying et al. (2019)*.

These and many other research questions lead to a vast amount of open research directions, which will benefit the field and lead to many applications in other computer science domains.

In our study, we focus on another interesting question regarding the fact that there are almost no general studies that compare the performance of models based on graph properties, most of the models are created for specific graph use-case. Below, we provide our insights on real-world networks as well as interpretations on such findings.

## MODEL COMPARISON

This paper focuses on the four most popular tasks on graphs: node classification, link prediction, node clustering and network visualization. These tasks cover most of the real-world applications, in which a graph is used to unify information on nodes and their properties.

### Data

We use four benchmark datasets for comparison of different models: CORA (*Sen et al., 2008*), Citeseer (*Lim & Buntine, 2016*), HSE coauthor network (*Makarov et al., 2018*), and Microsof Academic Graph Computer Science (MAG CS) (*Sinha et al., 2015*).

First two datasets are citation networks. This type of networks is very common for evaluating the quality of network embeddings. Also, these datasets are convenient for comparison of models, because they have interesting label and feature structure. The last dataset is a co-authorship network. It has heterogeneous edges and large size. General puspose graph embedding models work only with homogeneous graphs, so we merge all the edges between nodes in one edge. A brief overview of the datasets statistics is provided in Table 2.

**Table 2 Datasets description.**

|  | Assortativity | Label modularity | #Nodes | #Edges | #Features | #Classes |
|---|---|---|---|---|---|---|
| CORA | 0.7711 | 0.8061 | 2708 | 10,556 | 1,433 | 7 |
| CITESEER | 0.6754 | 0.8872 | 3327 | 9,228 | 3,703 | 6 |
| HSE | – | – | 4181 | 12,004 | 0 | 0 |
| MAG CS | 0.7845 | 0.6989 | 18333 | 16,3788 | 6,805 | 15 |

## Metrics

We use standard classification metrics for node classification and link prediction tasks.

- Accuracy is the rate of the right answer of a classifier.
- Precision is the rate of true positive answers relative to the number of all positive answers of a classifier.
- Recall is the rate of true positive answers relative to the number of all positive examples in the data.
- F1 is a harmonic mean of precision and recall.
- Area Under ROC-curve shows the probability that a random negative example sampled from a uniform distribution is ranked lower than randomly sampled positive.
- Average precision is the average of all possible precision values weighted by the recall for different probability thresholds.

We calculate the standard deviation with the following procedure:

1. Generate subsample of data with 90% volume of a given dataset.
2. Train model on it
3. Estimate quality of the trained model on the test set.
4. Repeat previous steps nine more times.

Described bootstrap (*Efron (1992)*) procedure allows to easily calculate standard error and confidence intervals for any statistics. Confidence intervals are required to understand the significance of the difference between models.

Node clustering was evaluated with two metrics: silhouette coefficient and modularity.

- Silhouette score shows the average similarity between each example and its cluster in comparison with the closest another cluster, so it measures the overall cluster separation relative to the distance measure. In our study, we use the Euclidean distance.
- Modularity score works pretty similarly but for computing inter- and intra-cluster quality, it measures the density of connections between clusters, respectively to its density inside clusters.

We also evaluate the quality of node clustering and network visualization with a visual comparison of how clusters are grouped in the UMAP (*McInnes, Healy & Melville, 2018*) projection of embeddings. UMAP (Uniform Manifold Approximation and Projection)

is the dimensionality reduction technique based on topology and Riemannian geometry. Firstly, it builds the weighted nearest neighbors graph according to elements feature vectors (embeddings). Then, it initializes layout using spectral embedding and optimizes it using SGD minimizing fuzzy-set cross-entropy.

The UMAP works much faster then TSNE and gives at least the same quality of the projection. Interpretation of received plot is simple: similar samples in initial space (for e.g., nodes with the same labels) should lie closely in the 2D plane.

## Evaluation pipeline

The node classification task is native multi-class classification. Link prediction task can be also solved as classification, but with two classes depicting edge existence. The basic approach on validating such methods is to use delayed sample. So, before any model was trained, we create a train-test split for all the datasets, in order to compare all the models on similar subsets. We use simple 50% test, 50% train random split for node classification, following other papers on graph embeddings.

The problem with link prediction is that described graphs are high-imbalanced because there are much more unconnected node pairs. Large imbalance leads to poor training because even simple constant prediction will give high-scores. One of the methods for working with this problem is to the under-sample larger class. To keep the classification task harder, it is convenient to use a negative sampling technique. We select the most similar pairs of nodes which are not connected in the same amount as existent edges. The used proximity metric is cosine similarity, which is a normalized dot product of feature vectors. For features, we use the adjacency matrix.

Because basic classification models do not work with discrete graph data, after developing train and test samples, we need to generate meaningful features. Here we use the unsupervised graph embeddings (128 dimensions as commonly used in different papers and surveys). Graph neural networks were also trained in an unsupervised way with reconstruction loss over the graph adjacency matrix. Reconstruction loss calculates with binary cross-entropy between adjacency matrix and its estimation achieved by inner-product decoding from the embedding matrix.

Now, we can solve downstream tasks like classification or clustering. For that part, we use three different classifiers: logistic regression (LR), random forest (RF) and gradient boosting (GBM). Logistic regression is a linear model: it calculates the weighted average of object features and normalizes it with sigmoid to receive probability. Linear models are fast, interpretable and easily tunable because of their simplicity. Random forest is the ensemble of decision trees built on bootstrapped subsamples in both dimensions features and observations. Gradient boosted machines are another approach to learn decision tree ensemble. It determines each next tree by sequential optimization of the previous learner error-term. The main advantage of the tree-based models is that they could recover non-linear dependencies. But for this flexibility, we pay with a strong dependance on hyperparameter selection. Scikit-learn implementation with default hyperparameters was used. In the link prediction, we simply concatenate node vectors to achieve edge embedding. For the clustering, we use the K-means model from Scikit-learn.

*Remark.* The common way to use graph neural networks is semi-supervised training. Such an approach gives a bias towards the usage of that model, because embedding learns not only graph structure and feature transformations, but supervised information about labels on the other hand. So we train graph neural networks in an unsupervised way because our study is aimed to understand how different properties of embedding models can help in considered tasks.

## Models

We select several models of different types mentioned in "Methods for Constructing Graph Embedding" that preserve different properties of a graph. The general idea is to compare models of different fitting approaches with respect to network properties.

Matrix factorization based models:

- GraRep is symmetric and preserves high-proximity. The default $K$-hop order is 5, the number of SVD iterations is 20, a random seed is 42.
- HOPE directly models asymmetric similarities.
- M-NMF preserves community structure. The default number of clusters is 20, clustering penalty is 0.05, modularity regularization penalty is 0.05, similarity mixing parameter is 5, the number of power-iterations is 200, early stopping step is 3.

Random-walks based models:

- Node2vec is a baseline for sequential methods which efficiently trade-offs between different proximity levels. The default walk length is 80, the number of walks per node is 10, return hyper-parameter is 1, in-out hyper-parameter is 1.
- Diff2vec use diffusion to sample random walks. The default number of nodes per diffusion tree is 80, the number of diffusions per source node is 10, context-size is 10, the number of ASGD iterations is 1, the learning rate is 0.025.
- Walklets allow to model different levels of community structure and generalize GraRep model. Default random walk length is 80, the number of random walks per source node is 5, the window size is 5, the minimal number of appearances is 1, the order of random walk is first, return hyper-parameter is 1, in-out hyper-parameter is 1.
- GEMSEC directly cluster nodes. Default random walk length is 80, the number of random walks per source node is 5, the window size is 5, the minimal number of appearances is 1, the order of random walk is first, return hyper-parameter is 1, in-out hyper-parameter is 1, distortion is 0.75, negative samples number is 10, the initial learning rate is 0.001, annealing factor for learning rate is 1, initial clustering weight coefficient is 0.1, final clustering weight coefficient is 0.5, smoothness regularization penalty is 0.0625, the number of clusters is 20, normalized overlap weight regularization.

Deep learning models:

- GCN is a baseline for deep learning models. The default number of epochs is 200, dropout is 0.3, the learning rate is 0.01, weight decay is 0.0005, the number of hidden layers is 1.

- GraphSage (GS) improves GCN by reducing the number of neighbors while weighting the node vectors. The dropout is 0.1, aggregation is GCN, the number of epochs is 200, the learning rate is 0.01, weight decay is 0.0005.
- GAT utilizes an attention mechanism. The number of epochs is 200, in-dropout is 0.1, attention dropout is 0.1, the learning rate is 0.005, the negative slope is 0.2, weight decay is 0.0005, the number of hidden layers is 1.

## RESULTS

The current section has the following structure. We start the analysis from node clustering tasks because it also helps to understand the performance of graph embeddings on the other tasks. Further, we describe node classification task and link prediction followed by network visualization. We also conducted experiments on random graphs to study the difference of graph embeddings on real-world networks and simulated ones.

### Node clustering

The results on the node clustering task are presented in Table 3. Rows depict different models, which are grouped by model type: matrix factorization, random walks, graph neural networks with and without features. On the columns, we can see results on different datasets. For each dataset, we calculate two metrics: modularity and silhouette score. Highlighted results are the best.

In node clustering task, results are pretty obvious: the embeddings, which work with community structure, perform the best in terms of modularity. GEMSEC directly penalizes embeddings for low modularity score with $K$-Means objective, Walklets catches this information by accounting for several levels of node neighborhood. Importance of such information could be proven by the comparatively high value of GraRep model, that works pretty similar to Walklets.

Graph neural networks with features give comparatively better results, meaning that node content helps to describe graph structure. GraphSAGE and GAT efficiently utilize the local structure of the network. The main difference is that GAT aggregates over the entire neighborhood, but GraphSAGE aggregates only over a fix-sized sample.

In the case of MAG CS graph (Table 4) the best results show GAT and GCN. It means that in the case of large, heterogeneous graph features play a major role. Interesting, that GAT without features works much better than other structural models. It could refer to the attention, that selects only the most important neighbors in node embedding construction. It seems that the attention mechanism helps in this case to distinguish heterogeneous edge nature.

GNN models trained in unsupervised fashion give poor results because they highly rely on the features when constructing embeddings even for learning graph structure.

The clustering results show that specific losses can dramatically increase quality on a specific task. As we will see further, such losses are also helpful in the node classification task preserving important graph properties.

**Table 3 Results of model validation on node clustering task (both metrics lie between (−1,1) and higher value means better results).** Bold corresponds to the best metric for each dataset.

| | CORA | | CITESEER | | HSE | |
|---|---|---|---|---|---|---|
| | Modularity | Silhouette | Modularity | Silhouette | Modularity | Silhouette |
| GRAREP (*Cao, Lu & Xu, 2015*) | 0.2249 | 0.1902 | 0.0320 | 0.3159 | 0.2320 | 0.3163 |
| HOPE (*Ou et al., 2016*) | 0.1222 | 0.2593 | 0.1748 | 0.5492 | 0.0027 | 0.6684 |
| NODE2VEC (*Grover & Leskovec, 2016*) | 0.0106 | 0.1000 | −0.0018 | 0.0464 | 0.0419 | 0.5576 |
| DIFF2VEC (*Rozemberczki & Sarkar, 2018*) | 0.1289 | 0.5412 | 0.0292 | 0.5422 | 0.1155 | 0.5429 |
| GEMSEC (*Rozemberczki et al., 2018*) | **0.7684** | 0.2280 | **0.7555** | 0.1508 | **0.7710** | 0.1678 |
| WALKLETS (*Perozzi, Kulkarni & Skiena, 2016*) | 0.7353 | 0.0812 | 0.7263 | 0.0566 | 0.7593 | 0.0667 |
| GCN *Kipf & Welling (2016a)* | 0.3800 | 0.3336 | 0.3754 | 0.4215 | – | – |
| GRAPHSAGE (*Hamilton, Ying & Leskovec, 2017a*) | 0.6455 | 0.3311 | 0.5774 | 0.4757 | – | – |
| GAT (*Veličković et al., 2017*) | 0.7209 | 0.3477 | 0.7367 | 0.3797 | – | – |
| GCN (NF) | −0.0096 | 0.3979 | 0.0360 | 0.4999 | 0.0008 | 0.6837 |
| GRAPHSAGE (NF) | 0.0212 | **0.7672** | 0.0960 | **0.9442** | 0.0552 | **0.8381** |
| GAT (NF) | 0.1335 | 0.2001 | 0.2968 | 0.3641 | 0.1400 | 0.6390 |

**Table 4 Results of model validation on node clustering task for MAG-CS dataset (both metrics lie between (−1,1) and higher value means better results).**

| | HOPE | NODE2VEC | GRAREP | WALKLETS | GCN | GAT | GCN (NF) | GAT (NF) |
|---|---|---|---|---|---|---|---|---|
| Modularity | −0.0001 | −0.0037 | 0.0027 | 0.0025 | 0.3462 | 0.3446 | 0.0112 | 0.1951 |
| Silhouette | 0.6548 | 0.0771 | 0.2348 | 0.0441 | 0.2369 | −0.0261 | 0.4654 | 0.0411 |

## Node classification

The results on node classification task are presented in Table 5. Rows show different types of models and columns show different datasets. For each dataset, we calculate accuracy for three different models: gradient boosted machines, logistic regression and random forest. Highlighted results are the best.

Models that have good performance in node clustering task also show high score in node classification. Labels in given datasets show different topics of articles, as soon as usually authors are dedicated to specific topics, so natural communities are constructed within these labels. This can also be proven by high modularity and assortativity coefficients of label communities for all graphs. In the classification task, it is also important to have a good separation of clusters, that could be measured by the silhouette coefficient. We can see those models that keep both high modularity and high silhouette work better.

Linear models show the comparatively lower score, but for random walk based embeddings, this difference is much less severe. Most of considered random walk models are based on Skip-Gram approach, which is a log-linear model. It reduces expression quality of the model but allows to learn vectors that perform well in linear models.

Results for MAG CS are presented in Table 6. Firstly, we compare fewer models, because we were not able to compute some embeddings for such a large graph, so we choose the

**Table 5  Results of model validation on node classification task (accuracy metric lies between (0,1) and higher value means better results).** Bold corresponds to the best metric for each dataset.

|  | GBM | LR | RF |
|---|---|---|---|
| *CORA* | | | |
| GRAREP (*Cao, Lu & Xu, 2015*) | 0.7610 ± 0.0434 | 0.7503 ± 0.0323 | 0.7751 ± 0.0254 |
| HOPE (*Ou et al., 2016*) | 0.7518 ± 0.0333 | 0.3024 ± 0.0308 | 0.7614 ± 0.0289 |
| M-NMF (*Wang et al., 2017d*) | 0.2596 ± 0.0250 | 0.2799 ± 0.0324 | 0.2633 ± 0.0239 |
| NODE2VEC (*Grover & Leskovec, 2016*) | 0.2522 ± 0.0200 | 0.2441 ± 0.0273 | 0.2441 ± 0.0257 |
| DIFF2VEC (*Rozemberczki & Sarkar, 2018*) | 0.2212 ± 0.0635 | 0.2843 ± 0.0387 | 0.2500 ± 0.0293 |
| GEMSEC (*Rozemberczki et al., 2018*) | 0.8338 ± 0.0326 | 0.8153 ± 0.0390 | **0.8634 ± 0.0251** |
| WALKLETS (*Perozzi, Kulkarni & Skiena, 2016*) | 0.8142 ± 0.0252 | 0.8124 ± 0.0317 | 0.8327 ± 0.0326 |
| GCN (*Kipf & Welling, 2016a*) | 0.7803 ± 0.0378 | 0.6588 ± 0.0448 | 0.7718 ± 0.0380 |
| GRAPHSAGE (*Hamilton, Ying & Leskovec, 2017a*) | 0.8083 ± 0.0358 | 0.7385 ± 0.0391 | 0.8168 ± 0.0316 |
| GAT (*Veličković et al., 2017*) | 0.8194 ± 0.0304 | 0.7455 ± 0.0420 | 0.8264 ± 0.0324 |
| GCN (NF) | 0.3021 ± 0.0204 | 0.2969 ± 0.0238 | 0.2888 ± 0.0194 |
| GRAPHSAGE (NF) | 0.3017 ± 0.0298 | 0.3021 ± 0.0305 | 0.3017 ± 0.0298 |
| GAT (NF) | 0.3021 ± 0.0305 | 0.3021 ± 0.0305 | 0.3021 ± 0.0305 |
| *CITESEER* | | | |
| GRAREP (*Cao, Lu & Xu, 2015*) | 0.5582 ± 0.0577 | 0.5110 ± 0.0443 | 0.5834 ± 0.0453 |
| HOPE (*Ou et al., 2016*) | 0.5468 ± 0.0346 | 0.2663 ± 0.0443 | 0.5489 ± 0.0378 |
| M-NMF (*Wang et al., 2017d*) | 0.1767 ± 0.0220 | 0.1978 ± 0.0241 | 0.1909 ± 0.0311 |
| NODE2VEC (*Grover & Leskovec, 2016*) | 0.1815 ± 0.0253 | 0.1806 ± 0.0165 | 0.1867 ± 0.0237 |
| DIFF2VEC (*Rozemberczki & Sarkar, 2018*) | 0.2035 ± 0.0373 | 0.2239 ± 0.0281 | 0.1930 ± 0.0287 |
| GEMSEC (*Rozemberczki et al., 2018*) | 0.6754 ± 0.0343 | 0.5867 ± 0.0427 | **0.7175 ± 0.0247** |
| WALKLETS (*Perozzi, Kulkarni & Skiena, 2016*) | 0.6291 ± 0.0280 | 0.6243 ± 0.0228 | 0.6480 ± 0.0277 |
| GCN (*Kipf & Welling, 2016a*) | 0.6312 ± 0.0210 | 0.5092 ± 0.0272 | 0.6342 ± 0.0209 |
| GRAPHSAGE (*Hamilton, Ying & Leskovec, 2017a*) | 0.6450 ± 0.0228 | 0.5425 ± 0.0192 | 0.6586 ± 0.0309 |
| GAT (*Veličković et al., 2017*) | 0.6733 ± 0.0238 | 0.5582 ± 0.0443 | 0.6763 ± 0.0220 |
| GCN (NF) | 0.2729 ± 0.0272 | 0.2317 ± 0.0336 | 0.2792 ± 0.0260 |
| GRAPHSAGE (NF) | 0.1996 ± 0.0409 | 0.1996 ± 0.0409 | 0.1996 ± 0.0409 |
| GAT (NF) | 0.1996 ± 0.0409 | 0.1996 ± 0.0409 | 0.1996 ± 0.0409 |

fastest ones with good performance in other experiments. HOPE outperforms other classic non-attributed embeddings. It is also interesting that the linear mixture of embedding elements works much better for this embedding then others. One of the reasons for such behavior is that in the large graphs, embeddings could be noisy and simple models could give better quality. It could be also a reason for the worse performance of $K$-hop based embeddings (Walklets and GraRep). But the main driver in node classification quality is the features of nodes. It could be seen from the high results of GCN and GAT models.

HOPE has a dramatic difference between linear and non-linear models because it estimates Katz centrality, which has non-linear nature. Also, we use HOPE implementation from GEM *Goyal & Ferrara (2017)*, where node embedding is achieved as a concatenation of its self- and context-representations. The non-linear model helps to reconstruct the dot

**Table 6 Results of model validation on node classification task for MAG-CS dataset (accuracy metric lies between (0,1) and higher value means better results).** Bold corresponds to the best metric for each dataset.

|  | GBM | LR | RF |
| --- | --- | --- | --- |
| GRAREP (*Cao, Lu & Xu, 2015*) | 0.1915 ± 0.0162 | 0.1404 ± 0.0217 | 0.1737 ± 0.0169 |
| HOPE (*Ou et al., 2016*) | 0.1985 ± 0.0233 | 0.2255 ± 0.021 | 0.1665 ± 0.0184 |
| NODE2VEC (*Grover & Leskovec, 2016*) | 0.1882 ± 0.034 | 0.2048 ± 0.0332 | 0.168 ± 0.0195 |
| WALKLETS (*Perozzi, Kulkarni & Skiena, 2016*) | 0.1866 ± 0.0171 | 0.1527 ± 0.0084 | 0.1886 ± 0.0189 |
| GCN (*Kipf & Welling, 2016a*) | 0.752 ± 0.0177 | 0.7317 ± 0.0166 | **0.7568 ± 0.0176** |
| GAT (*Veličković et al., 2017*) | 0.6272 ± 0.0188 | 0.6424 ± 0.0202 | 0.6095 ± 0.0208 |
| GCN (NF) | 0.2089 ± 0.0154 | 0.2255 ± 0.021 | 0.1738 ± 0.0117 |
| GAT (NF) | 0.2255 ± 0.021 | 0.2255 ± 0.021 | 0.2255 ± 0.021 |

product decoder. A similar argument can explain diversity in neural network models, but it has less variance because of high clustering efficiency.

## Link prediction

Table 7 shows results for link prediction task. It is separated into three groups by datasets. Rows represent different graph embedding models and columns show different second-level classification models: gradient boosted machines, logistic regression and random forest. Highlighted results are the best.

In the link prediction task, we can also see the importance of clustering. Links are more likely to occur within one community. The high-proximity models also work much better, because in that task we need to understand how similar are non-connected nodes.

The performance of the HOPE model in this task is more significant. HOPE model concentrates on preserving asymmetric transitivity. The older paper can not cite the newer one.

GCN without features performs much better than other graph neural networks. It accounts for the whole neighborhood and directly uses the adjacency matrix to train embeddings.

Results for MAG CS (Table 8) are consistent with these findings. However, despite the good quality on the community clustering task, GAT without features shows pure performance on the Link prediction task. However, GCN without features is close to the GAT with features. It means that in this task it is necessary to account the whole neighborhood.

A dramatic difference in the quality of linear and non-linear models can be explained by the objective of the link prediction task. It requires to model the proximity between to nodes. Such metrics are non-linear. So for reconstructing it from concatenated vectors of nodes, we need some non-linear transformations.

## Graph visualization

We present results of node clustering jointly with network visualization using UMAP technique. The results for three different datasets are shown at Fig. 1 for Cora, Fig. 2 for Citeseer, and Fig. 3 for HSE datasets, respectively.

**Table 7 Results of model validation on link prediction task (accuracy metric lies between (0,1) and higher value means better results).** Bold corresponds to the best metric for each dataset.

| | GBM | LR | RF |
|---|---|---|---|
| *CORA* | | | |
| GRAREP (*Cao, Lu & Xu, 2015*) | 0.8766 ± 0.0056 | 0.7585 ± 0.0037 | 0.9143 ± 0.0021 |
| HOPE (*Ou et al., 2016*) | 0.9422 ± 0.0039 | 0.6706 ± 0.0032 | 0.9478 ± 0.0020 |
| M-NMF (*Wang et al., 2017d*) | 0.6507 ± 0.0038 | 0.6252 ± 0.0018 | 0.6618 ± 0.0022 |
| NODE2VEC (*Grover & Leskovec, 2016*) | 0.7047 ± 0.0039 | 0.6185 ± 0.0037 | 0.7060 ± 0.0042 |
| DIFF2VEC (*Rozemberczki & Sarkar, 2018*) | 0.7780 ± 0.0049 | 0.7508 ± 0.0045 | 0.7413 ± 0.0029 |
| GEMSEC (*Rozemberczki et al., 2018*) | **0.9692 ± 0.0030** | 0.6512 ± 0.0052 | 0.9653 ± 0.0011 |
| WALKLETS (*Perozzi, Kulkarni & Skiena, 2016*) | 0.9153 ± 0.0058 | 0.7073 ± 0.0022 | 0.9574 ± 0.0017 |
| GCN (*Kipf & Welling (2016a*) | 0.8784 ± 0.0028 | 0.7094 ± 0.0041 | 0.8978 ± 0.0022 |
| GRAPHSAGE (*Hamilton, Ying & Leskovec, 2017a*) | 0.8988 ± 0.0050 | 0.5668 ± 0.0049 | 0.9111 ± 0.0028 |
| GAT (*Veličković et al., 2017*) | 0.9127 ± 0.0047 | 0.5666 ± 0.0063 | 0.9337 ± 0.0021 |
| GCN (NF) | 0.7852 ± 0.0060 | 0.7084 ± 0.0033 | 0.8014 ± 0.0024 |
| GRAPHSAGE (NF) | 0.5459 ± 0.0043 | 0.5033 ± 0.0021 | 0.5459 ± 0.0043 |
| GAT (NF) | 0.5033 ± 0.0021 | 0.5033 ± 0.0021 | 0.5033 ± 0.0021 |
| *CITESEER* | | | |
| GRAREP (*Cao, Lu & Xu, 2015*) | 0.8786 ± 0.0046 | 0.7198 ± 0.0049 | 0.9254 ± 0.0031 |
| HOPE (*Ou et al., 2016*) | 0.8985 ± 0.0074 | 0.6358 ± 0.0052 | 0.9119 ± 0.0029 |
| M-NMF (*Wang et al., 2017d*) | 0.5926 ± 0.0049 | 0.5685 ± 0.0033 | 0.6215 ± 0.0031 |
| NODE2VEC (*Grover & Leskovec, 2016*) | 0.6895 ± 0.0050 | 0.6315 ± 0.0056 | 0.6934 ± 0.0046 |
| DIFF2VEC (*Rozemberczki & Sarkar, 2018*) | 0.7553 ± 0.0038 | 0.7258 ± 0.0038 | 0.7206 ± 0.0060 |
| GEMSEC (*Rozemberczki et al., 2018*) | **0.9827 ± 0.0031** | 0.6151 ± 0.0096 | 0.9726 ± 0.0026 |
| WALKLETS (*Perozzi, Kulkarni & Skiena, 2016*) | 0.8688 ± 0.0066 | 0.6672 ± 0.0040 | 0.9429 ± 0.0024 |
| GCN (*Kipf & Welling, 2016a*) | 0.8863 ± 0.0033 | 0.6910 ± 0.0032 | 0.9052 ± 0.0024 |
| GRAPHSAGE (*Hamilton, Ying & Leskovec, 2017a*) | 0.8952 ± 0.0037 | 0.6082 ± 0.0036 | 0.8998 ± 0.0034 |
| GAT (*Veličković et al., 2017*) | 0.9175 ± 0.0030 | 0.6136 ± 0.0051 | 0.9306 ± 0.0025 |
| GCN (NF) | 0.7892 ± 0.0039 | 0.6881 ± 0.0044 | 0.8100 ± 0.0034 |
| GRAPHSAGE (NF) | 0.5181 ± 0.0039 | 0.5037 ± 0.0026 | 0.5181 ± 0.0039 |
| GAT (NF) | 0.5037 ± 0.0026 | 0.5037 ± 0.0026 | 0.5037 ± 0.0026 |
| *HSE* | | | |
| GRAREP (*Cao, Lu & Xu, 2015*) | 0.9202 ± 0.0068 | 0.7956 ± 0.0032 | 0.9332 ± 0.0022 |
| HOPE (*Ou et al., 2016*) | 0.6590 ± 0.0050 | 0.6062 ± 0.0055 | 0.7022 ± 0.0038 |
| M-NMF (*Wang et al. (2017d*) | 0.6824 ± 0.0058 | 0.6277 ± 0.0041 | 0.7467 ± 0.0032 |
| NODE2VEC (*Grover & Leskovec, 2016*) | 0.7257 ± 0.0049 | 0.6634 ± 0.0034 | 0.7592 ± 0.0039 |
| DIFF2VEC (*Rozemberczki & Sarkar, 2018*) | 0.7850 ± 0.0040 | 0.7505 ± 0.0037 | 0.7795 ± 0.0034 |
| GEMSEC (*Rozemberczki et al., 2018*) | **0.9724 ± 0.0035** | 0.7065 ± 0.0043 | 0.9671 ± 0.0013 |
| WALKLETS (*Perozzi, Kulkarni & Skiena, 2016*) | 0.9484 ± 0.0028 | 0.7730 ± 0.0035 | 0.9615 ± 0.0022 |
| GCN (NF) (*Kipf & Welling, 2016a*) | 0.8178 ± 0.0021 | 0.7867 ± 0.0031 | 0.8214 ± 0.0030 |
| GRAPHSAGE (NF) (*Hamilton, Ying & Leskovec, 2017a*) | 0.5071 ± 0.0026 | 0.5039 ± 0.0030 | 0.5071 ± 0.0026 |
| GAT (NF) (*Veličković et al., 2017*) | 0.5039 ± 0.0030 | 0.5039 ± 0.0030 | 0.5039 ± 0.0030 |

**Table 8  Results of model validation on link prediction task for MAG-CS dataset (accuracy metric lies between (0,1) and higher value means better results).** Bold corresponds to the best metric for each dataset.

|  | GBM | LR | RF |
|---|---|---|---|
| GRAREP (*Cao, Lu & Xu, 2015*) | 0.5986 ± 0.0047 | 0.5626 ± 0.0016 | 0.5998 ± 0.0025 |
| HOPE (*Ou et al., 2016*) | 0.566 ± 0.0017 | 0.5275 ± 0.0025 | 0.6007 ± 0.0027 |
| NODE2VEC (*Grover & Leskovec, 2016*) | 0.578 ± 0.0023 | 0.5425 ± 0.0015 | 0.6137 ± 0.0031 |
| WALKLETS (*Perozzi, Kulkarni & Skiena, 2016*) | 0.5798 ± 0.0024 | 0.5647 ± 0.0015 | 0.6077 ± 0.0027 |
| GCN (*Kipf & Welling, 2016a*) | 0.8486 ± 0.0022 | 0.6553 ± 0.0014 | **0.8772 ± 0.0012** |
| GAT (*Veličković et al., 2017*) | 0.7293 ± 0.0041 | 0.5632 ± 0.0024 | 0.7524 ± 0.0027 |
| GCN (NF) | 0.7253 ± 0.0012 | 0.697 ± 0.0014 | 0.7261 ± 0.0017 |
| GAT (NF) | 0.5015 ± 0.0009 | 0.5015 ± 0.0009 | 0.5015 ± 0.0009 |

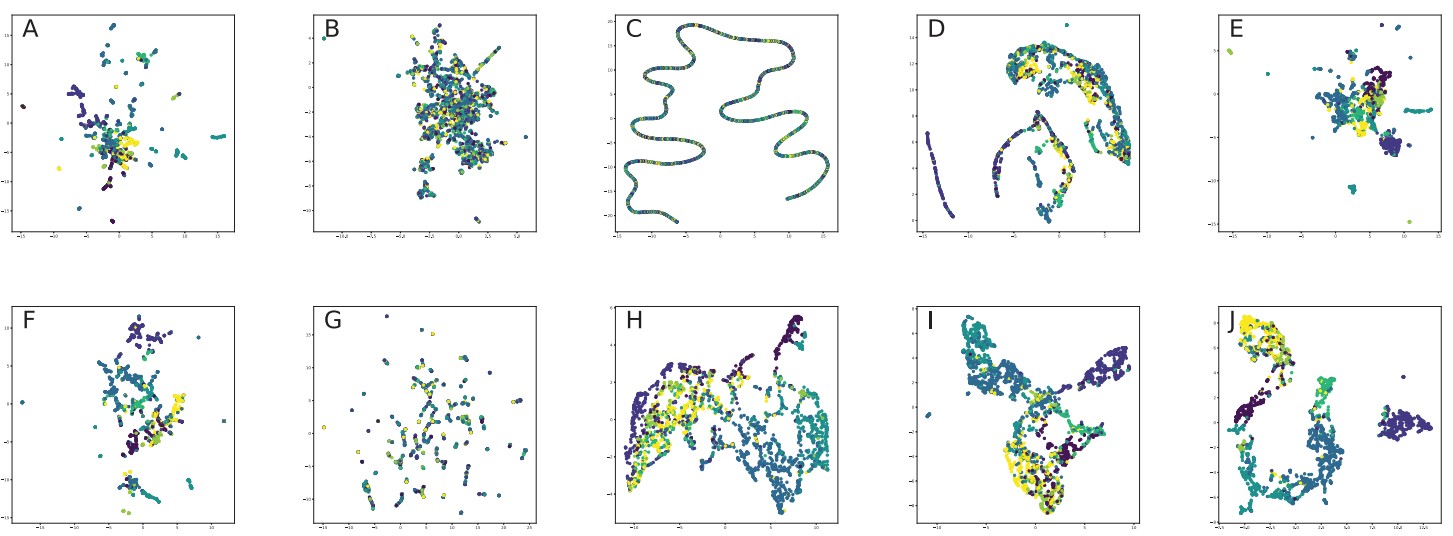

**Figure 1  UMAP projection of CORA embeddings: (A) HOPE. (B) Node2Vec. (C) Diff2Vec. (D) GraRep. (E) Walklets. (F) GEMSEC. (G) M-NMF. (H) GCN. (I) GraphSage. (J) GAT.**

The best visualization in terms of community structure seems to be Walklets and GraRep models, which give nicely distinguishable clusters in all the cases. Both models work in the same way with $k$-hop similarity of vertices. GEMSEC also provides separate cluster picture but creates a lot of noisy points.

Interestingly, that HOPE also split graphs into several parts, but we can see by the modularity score, such parts are not correlated with node communities. Such an effect could occur because HOPE embedding has non-homogeneous structure due to concatenation of self- and context-representations.

In the case of graph neural networks, except for GAT, all clusters have poor separation. Such effect occurs because GNN weights neighborhood node attributes, so boundary nodes will be close. GAT allows mitigating this problem because utilizes the attention mechanism, which weights meaningless node neighbors to zero.

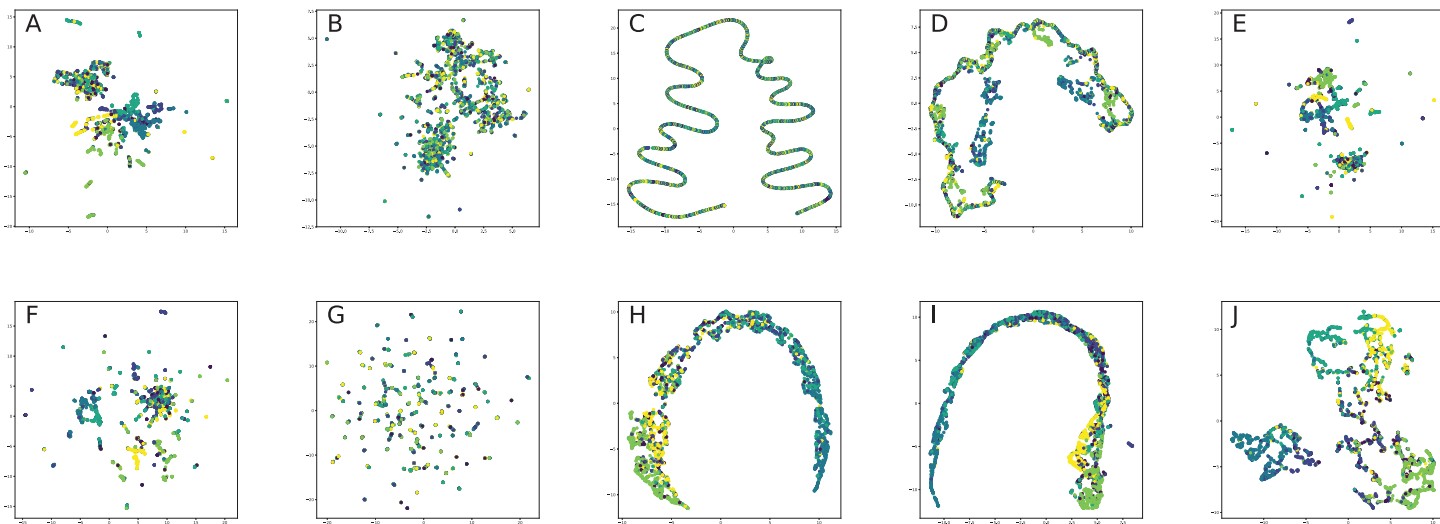

**Figure 2** UMAP projection of Citeseer embeddings: (A) HOPE. (B) Node2Vec. (C) Diff2Vec. (D) GraRep. (E) Walklets. (F) GEMSEC. (G) M-NMF. (H) GCN. (I) GraphSage. (J) GAT.

It seems that one of the most important graph property in the studied tasks is the community structure. So, most of the methods, that works with it directly allow to achieve the best scores. It is also connected to the level of proximity because it is indirectly connected with the community structure. The graph neural networks allow to easily catch node attributes, but miss most of graph properties information, so it performs on the level of baseline models without it.

## Random graphs

In order to understand robustness of graph embeddings, we decided to test how modeling real-world network with random graphs impact on the quality of graph embeddings for simulated networks.

Firstly, we should explain the formation of random graphs in different models. *Erdös & Rényi (1959)* builds the graphs using a simple binomial rule for creating an edge with a given density of graph. *Barabási & Albert (1999)* starts from a small graph and sequentially adds a new node with a given density and connects existing nodes using preferential attachment rule. In the *Watts & Strogatz (1998)* model, the regular lattice is firstly constructed followed by edge rewiring procedure.

We build random graphs regarding the properties of real-world graphs. To build ER graph one need to have a number of nodes and edges in the graph. For the BA graph construction, it is required to have a number of nodes and number of edges for the newly added node at each iteration. It is a small integer, so we just select the best to fit the number of edges of benchmarks. The parameters of WS graphs were chosen based on the number of nodes, edges and average clustering of graphs following formulae: the number of edges in starting lattice is equal to $k = [2\# \text{ edges} \# \text{ nodes}]$, the rewriting probability is equal to $p = 1 - \sqrt[3]{4(k-1)/3(k-2)} \cdot (average\ clustering)$ (*Barrat & Weigt, 2000*).

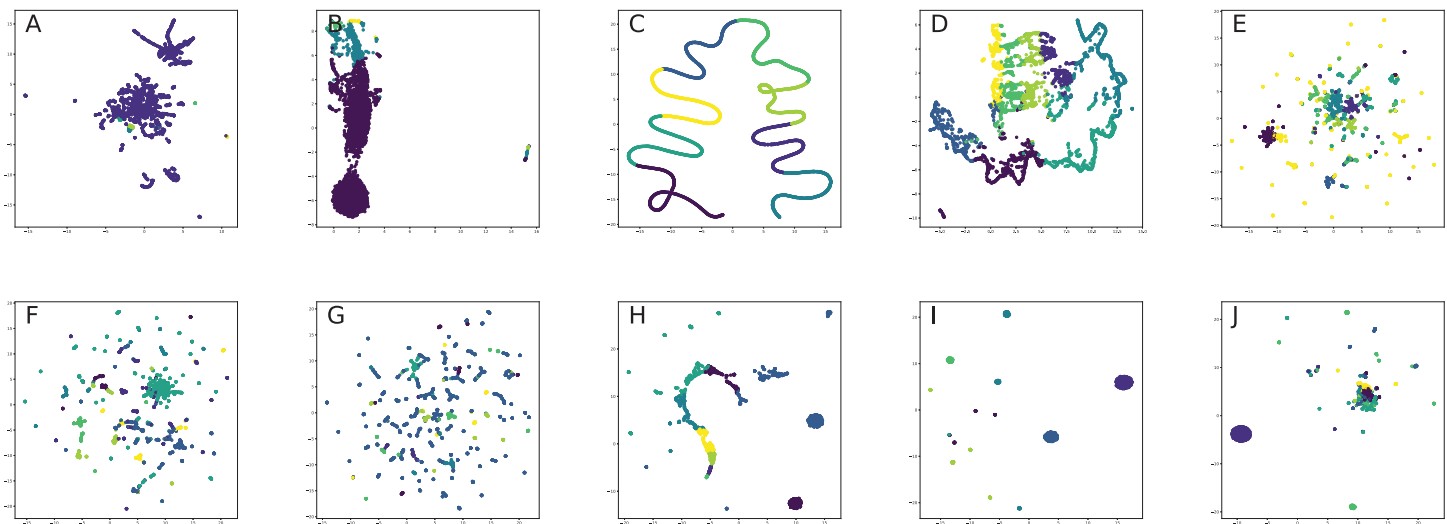

**Figure 3** UMAP projection of HSE embeddings: (A) HOPE. (B) Node2Vec. (C) Diff2Vec. (D) GraRep. (E) Walklets. (F) GEMSEC. (G) M-NMF. (H) GCN. (I) GraphSage. (J) GAT.

**Table 9 Results of model validation on node clustering task for random graphs (both metrics lie between (−1,1) and higher value means better results.** Bold corresponds to the best metric for each dataset.

|  | HOPE | | WALKLETS | |
| --- | --- | --- | --- | --- |
|  | Modularity | Silhouette | Modularity | Silhouette |
| CORA (Original) | 0.1222 | 0.2593 | 0.7353 | 0.0812 |
| CORA (Barabási-Albert) | 0.0005 | **0.1807** | 0.1465 | 0.0046 |
| CORA (Erdős-Rényi) | −0.0022 | 0.0216 | 0.0184 | 0.0080 |
| CORA (Watts-Strogatz) | 0.0629 | 0.1180 | **0.5251** | 0.0212 |
| CITESEER (Original) | 0.1748 | 0.5492 | 0.7263 | 0.0566 |
| CITESEER (Barabási-Albert) | −0.0008 | **0.3731** | 0.0397 | −0.0006 |
| CITESEER (Erdős-Rényi) | 0.0031 | 0.1495 | −0.0040 | 0.0085 |
| CITESEER (Watts-Strogatz) | 0.0344 | 0.1270 | **0.4941** | 0.0155 |

One of the main properties of all random graph models is the giant connected component. So embedding models learn it on the train part and works better than random in link prediction task in all cases. Additionally, in the BA model, there are some nodes with much larger density than others, so it is easier to predict missed links. Watts–Strogatz also has a straightforward mechanism of edge construction, where the shortest path is small. In both BA and WS models it is also possible to reproduce community structure. We can see it by large modularity metric in the clustering task.

Random graph modeling is one of the efficient methods in network science for evaluation of different model properties. For example, comparison of real-graph with its random analog could help to understand how good is the received quality for the specific task. We follow this idea and compare two embeddings models Walklets and HOPE.

**Table 10 Results of model validation on link prediction task for random graphs (accuracy metric lies between (0,1) and higher value means better results).** Bold corresponds to the best metric for each dataset.

| | RF | LR | GBM |
|---|---|---|---|
| *Walklets* | | | |
| CORA Original | 0.8142 ± 0.0252 | 0.8124 ± 0.0317 | 0.8327 ± 0.0326 |
| CORA-ER | 0.617 ± 0.0081 | 0.5658 ± 0.013 | 0.5904 ± 0.0141 |
| CORA-BA | 0.7216 ± 0.0113 | 0.6928 ± 0.0103 | 0.7271 ± 0.0136 |
| CORA-WS | 0.6511 ± 0.0329 | 0.5168 ± 0.0075 | **0.7442 ± 0.0762** |
| CITESEER Original | 0.6291 ± 0.0280 | 0.6243 ± 0.0228 | 0.6480 ± 0.0277 |
| CITESEER-ER | 0.5505 ± 0.0062 | 0.5335 ± 0.0071 | 0.5411 ± 0.0076 |
| CITESEER-BA | 0.6807 ± 0.0071 | 0.662 ± 0.0123 | **0.6871 ± 0.018** |
| CITESEER-WS | 0.571 ± 0.0142 | 0.5232 ± 0.022 | 0.6121 ± 0.031 |
| *HOPE* | | | |
| CORA Original | 0.7518 ± 0.0333 | 0.3024 ± 0.0308 | 0.7614 ± 0.0289 |
| CORA-ER | 0.5936 ± 0.0042 | 0.5114 ± 0.0055 | 0.5734 ± 0.0063 |
| CORA-BA | **0.6521 ± 0.0071** | 0.5559 ± 0.0144 | 0.6312 ± 0.007 |
| CORA-WS | 0.5115 ± 0.0048 | 0.51 ± 0.0052 | 0.5132 ± 0.0071 |
| CITESEER Original | 0.5468 ± 0.0346 | 0.2663 ± 0.0443 | 0.5489 ± 0.0378 |
| CITESEER-ER | 0.5509 ± 0.015 | 0.5066 ± 0.0029 | 0.5439 ± 0.01 |
| CITESEER-BA | **0.6304 ± 0.0116** | 0.5422 ± 0.0071 | 0.6096 ± 0.0056 |
| CITESEER-WS | 0.5169 ± 0.0057 | 0.5093 ± 0.0058 | 0.521 ± 0.0088 |

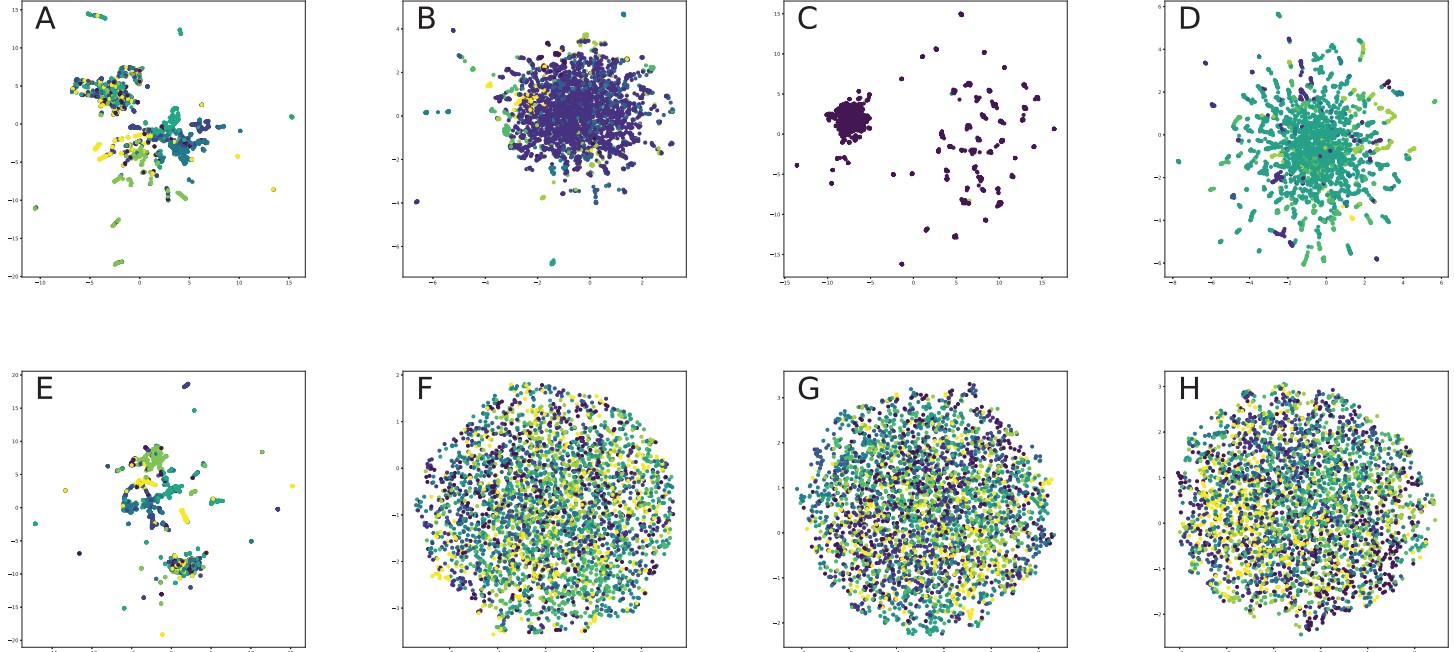

**Figure 4 UMAP projection of Citeseer based random graph embeddings: (A) Original graph, HOPE. (B) Erdős-Rényi, HOPE. (C) Barabási-Albert, HOPE. (D) Watts-Strogatz, HOPE. (E) Original graph, Walklets. (F) Erdős-Rényi, Walklets. (G) Barabási-Albert, Walklets. (H) Watts-Strogatz, Walklets.**

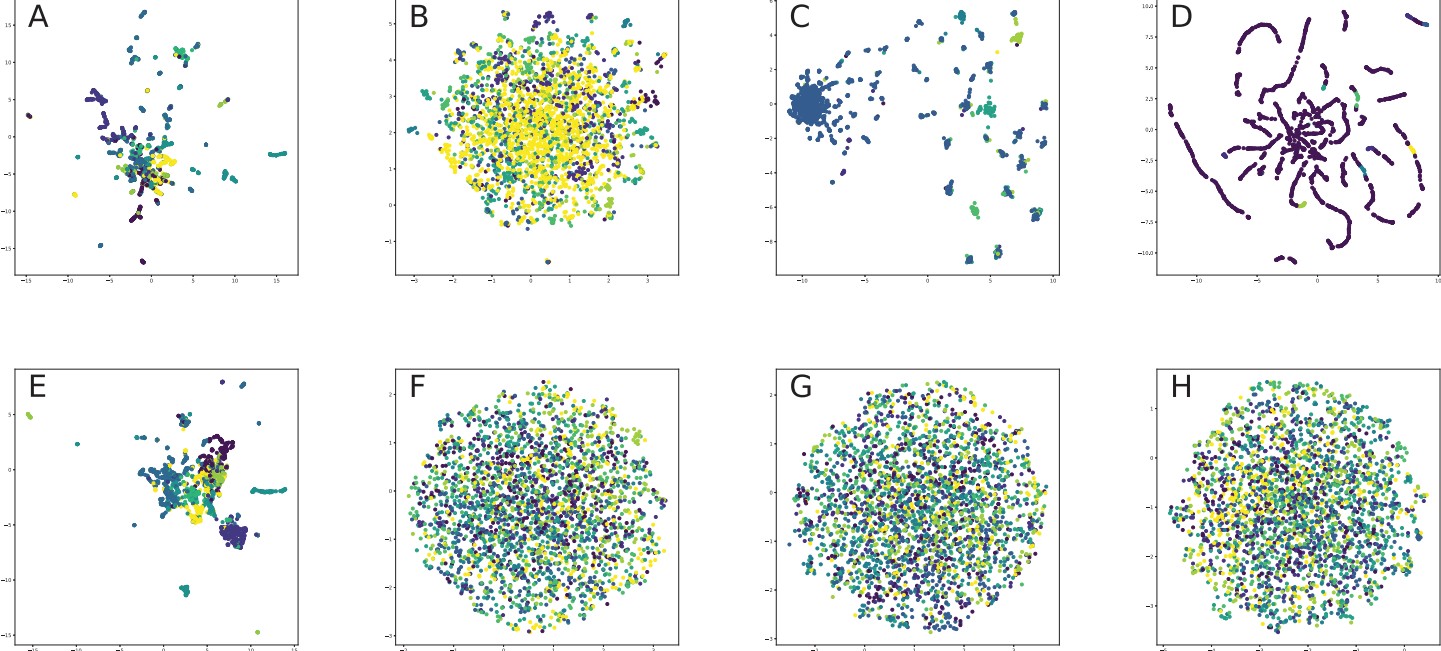

**Figure 5** UMAP projection of CORA based random graph embeddings: (A) Original graph, HOPE. (B) Erdős-Rényi, HOPE. (C) Barabási-Albert, HOPE. (D) Watts-Strogatz, HOPE. (E) Original graph, Walklets. (F) Erdős-Rényi, Walklets. (G) Barabási-Albert, Walklets. (H) Watts-Strogatz, Walklets.                              

We select these embeddings because it is non-context, show good performance and saves different properties. Walklets preserves *K*-hop similarity and HOPE preserves asymmetric transitivity.

Similarly to the experiments on the real-world graphs, Walklets shows superior performance in comparison to the HOPE in Clustering (Table 9) and LPP (Table 10) tasks.

However, visually (Figs. 4 and 5) it better separates dense clusters. The Walklets visualization of random graphs differs from real-world cases. Random graphs give much sparser and visually harder to distinguish structure.

The results on random graphs and real networks differ sufficiently. It means that embedding models could really learn graph structure and its properties. Also, such citation networks are poorly described by random graph models.

## CONCLUSION

In the current work, we present a comprehensive survey of graph embedding techniques. The work overviews different types of graph embeddings with respect to methods, network types, their applications to computer science domains.

One of the main achievements at the moment are the scalable models. The GNN could be trained in batch and distributed fashion. Such methods allow using powerful attribute-aware models for real-world large graphs. However, only a few works analyze the proper strategies for batch sampling and its effect on the final results in terms of bias-variance trade-off. Another way to accelerate GNN training is to coarse a graph, but it could affect

dramatically the final quality of the model. So, the understanding and further developing of coarsening and sampling techniques is the promising direction.

One of the most popular technique for graph embedding at the current time is the attention mechanism. It helps to account for some significant properties of a graph like temporality and heterogeneity by introducing attention in different dimensions: time, different levels of edge and nodes types. However, this method could be exhaustive in terms of computation, so it should be used with acceleration techniques.

The other research direction that grows rapidly is the stability of graph embeddings. The popular practices are to use variational graph autoencoders with Gaussian denoising or adversarial training.

The development of scalable and high-quality graph neural networks leads to an increase in the number of applications to the non-graph domain. The most common application of it is the modeling of similarity or nearest neighbors graphs. Such approaches are presented in natural language processing, computer vision and recommender systems. However, in many fields, structures could be natively presented as graphs in terms of labels (samples of one type are connected), interaction, knowledge or relation graphs.

Our survey covers the most complete of methods and application in different computer science domains related to machine learning problems on relational data.

In addition, in the experiment part of the study we provide results on training best graph embedding models for node classification, link prediction, node clustering and network visualization tasks for different types of models and graphs to understand why certain graph embedding perform better than others on benchmark datasets under different training settings. Our experiments explain how different embeddings work with different properties uncovering graph inner properties and descriptive statistics impact on the models performance. As one of the most interesting findings, we show that structural embeddings with proper objectives achieve competitive quality vs graph neural networks.

Still, it could be hard to apply such methods to large graphs. Firstly, there is a problem with high computational complexity. Graph neural networks solve this issue by using batch training and sampling techniques. Another problem is that learned structural embeddings for large graphs could be noisy. However, adding the node attributes helps to concentrate on the specific important properties. Modern models focus on accounting for node attributes, but it was found that more important question is how to balance a trade-off between node attributes and network structure. Our work will be helpful in the further development of such generalization methods to answer this question. Such methods will allow to easily apply graph models in different domains.

### Funding

The work of Nikita Nikitinsky on Section 6 was supported by the Russian Science Foundation grant 19-11-00281. The OA fee was covered under support of Faculty of Computer Science, HSE University. The funders had no role in study design, data collection and analysis, decision to publish, or preparation of the manuscript.

## Grant Disclosures

The following grant information was disclosed by the authors:

Russian Science Foundation: 19-11-00281.

Faculty of Computer Science, HSE University.

## Competing Interests

The authors declare that they have no competing interests.

## Author Contributions

- Ilya Makarov conceived and designed the experiments, analyzed the data, performed the computation work, prepared figures and/or tables, authored or reviewed drafts of the paper, and approved the final draft.
- Dmitrii Kiselev conceived and designed the experiments, performed the experiments, analyzed the data, performed the computation work, prepared figures and/or tables, authored or reviewed drafts of the paper, and approved the final draft.
- Nikita Nikitinsky conceived and designed the experiments, analyzed the data, authored or reviewed drafts of the paper, and approved the final draft.
- Lovro Subelj conceived and designed the experiments, analyzed the data, authored or reviewed drafts of the paper, and approved the final draft.

## Data Availability

Metric evaluation code for each of the presented tasks is available in the Supplemental Files.

## Supplemental Information

Supplemental information for this article can be found online at http://dx.doi.org/10.7717/peerj-cs.357#supplemental-information.

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
