# Peer review of "Survey on graph embeddings and their applications to machine learning problems on graphs"

_PeerJ Computer Science, doi:10.7717/peerj-cs.357_

## Round 0.1 · original submission · Major Revisions

This survey paper is interesting. However, there are disadvantages needing to be overcome based on the comments of the reviewers.

Reviewer 1 ·

Basic reporting

In this work, the author reviews several graph embedding methods from several perspectives. It is an interesting and popular field of research.

Experimental design

It is interesting to find some experiment in a survey paper. The author designs some interesting experiments to compare different embedding methods.

Validity of the findings

no comment

Additional comments

I have some high-level concerns over the paper. The paper is not well-organized and still needs some polish. It is not easy to follow sometimes. I suggest the authors spend some time polishing the paper. Some of them are listed below:
1) Introduction section lack pf some high-level paper structure explanation.
2) Section Preliminaries has no section number
3) The paper writing style is inconsistent in the paper.

·

Basic reporting

The writing of this paper is clear and easy to understand. The author provides relatively sufficient references, even though some pretty recent graph embedding techniques are not included. This is expected since current graph research community are blowing up.

I think the paper structure can be further improved. I feel like I was reading a tutorial, or some kind of literature reviews.

The result verify authors' claim. but the table 4 seems to be not completed (lacking last columns). Additionally, what is LR, RF means? What is GBM? for Table 3, 4, and 5, adding its corresponding references would be better. Without references, I am not sure how consistent with the original one.

Experimental design

Authors compare different methods on the examined dataset. However, compare with discussions before experiments, the experimental design seems to be limited regarding to the dataset: The author has mentioned multiple hypothesis, including heterogenous network, attributed network, variety of applications, network compression, etc. while the experimental only shows on CORA and CITESEER. These two citation dataset does not reflect network heterogeneity, and these dataset does not either consider attributes of each vertices.

Validity of the findings

Since this is another survey of graph embedding, I do expect authors would provide deeper insights among listed publications. Although the author claims that "Our survey is not limited to specific model types and provides an overview from different angles", I haven't found this survey deep dive existing pain points: For example, industrials often face very large graph so that the adjacent matrix cannot even fit into the memory, how to do graph embedding in such case; graph neural network is also facing over-smoothing problem as well; etc. I believe if such topics are surveyed, the contribution of this paper would be more significant.

Reviewer 3 ·

Basic reporting

There are a lot of grammatical errors and typos in this manuscript. The authors should re-check and revise carefully. For examples:
- ... a family of automated graph feature engineering techniques have been proposed in different streams of literature
- This paper focus on the four most popular tasks on graphs ...
- ...

How did the authors select the range of selecting papers? It needs to be described clearly.

There are some mistakes in article structure. For example, the authors have section "3.2.1", but no "3.2.2".

Deep neural networks have been used in previous works such as PMID: 31277574 and PMID: 32613242. Therefore, the authors should refer more works in this description to attract broader readership.

Experimental design

The authors used 3 benchmark datasets, but why did they only show 2 datasets in Table 2?

How did the authors tune the optimal hyperparameters of their deep learning models?

The authors should explain clearly on the UMAP idea as well as parameter setting for UMAP generation.

It is good to have an UMAP to analyze the clusters of the data, but why did the authors not include the UMAP information into the models to help improving the performance? The authors could show a comparison between with and without UMAP features.

The authors mentioned Walklets and GraRep models were better than the others. However, I have checked that the performance from the others were also good such as Diff2Vec, GCN, or GraphSage. Therefore, what is the metric to determine the performance of UMAP analyses?

Performance metrics (i.e., Recall, Precision, or Accuracy) have been used in previous works such as PMID: 31987913 and PMID: 28643394. It is suggested to refer more works here.

The authors should release source codes for replicating the results.

Validity of the findings

Discussions are weak. The authors should discuss more on the findings and comparisons to previous works.

Additional comments

No comment

---

## Round 0.2 · Minor Revisions

As the comments show, most issues have been solved. However, it still needs further improvement according to the suggestions of Reviewer 1.

Reviewer 1 ·

Basic reporting

I think the author has addressed all the comments from the reviewers properly. This paper is in better shape now. My only two concerns are:
1) reader may get confused about the paper structure after reading the introduction part. The author stated that they present "how different network properties affect graph embedding quality on most popular machine learning problems on graphs." It seems this paper focused on the graph network instead of the graph embedding techniques.
2) I suggest the author do proofread before publication.

Experimental design

no comment

Validity of the findings

no comment

Reviewer 3 ·

Basic reporting

No comment

Experimental design

No comment

Validity of the findings

No comment

Additional comments

My previous comments have been addressed satisfactorily.

---

## Round 0.3 · accepted · Accept

After reading the paper and the response letter, I believe the paper has been revised well.